# Silk fibroin hydrogel adhesive enables sealed-tight reconstruction of meniscus tears

Xihao Pan[1,2,3,4,8], Rui Li[1,3,8], Wenyue Li[1,2,3,8], Wei Sun[1,2], Yiyang Yan[1,2,3], Xiaochen Xiang[3], Jinghua Fang [5], Youguo Liao[1,2], Chang Xie[1,2], Xiaozhao Wang[1,2,3,4], Youzhi Cai[6], Xudong Yao[7] & Hongwei Ouyang [1,2,3,4] ✉

Despite orientationally variant tears of the meniscus, suture repair is the current clinical gold treatment. However, inaccessible tears in company with re-tears susceptibility remain unresolved. To extend meniscal repair tools from the perspective of adhesion and regeneration, we design a dual functional biologic-released bioadhesive (S-PIL10) comprised of methacrylated silk fibroin crosslinked with phenylboronic acid-ionic liquid loading with growth factor TGF-β1, which integrates chemo-mechanical restoration with inner meniscal regeneration. Supramolecular interactions of β-sheets and hydrogen bonds richened by phenylboronic acid-ionic liquid (PIL) result in enhanced wet adhesion, swelling resistance, and anti-fatigue capabilities, compared to neat silk fibroin gel. Besides, elimination of reactive oxygen species (ROS) by S-PIL10 further fortifies localized meniscus tear repair by affecting inflammatory microenvironment with dynamic borate ester bonds, and S-PIL10 continuously releases TGF-β1 for cell recruitment and bridging of defect edge. In vivo rabbit models functionally evidence the seamless and dense reconstruction of torn meniscus, verifying that the concept of meniscus adhesive is feasible and providing a promising revolutionary strategy for preclinical research to repair meniscus tears.

Meniscus injury is one of the most common injuries in the knee joint caused by acute sports trauma or age-associated degenerative changes[1,2]. Meniscus tears are usually unhealable as a result of the poor blood supply of the inner one-third of the meniscus (white–white zone)[3,4]. Tears not treated in time will bring about abnormal stress distribution in the knee, joint instability, cartilage degeneration, and eventually result in progressive osteoarthritis[5]. Partial or total meniscectomy is the most common surgical treatments for meniscus tears, while they only relieve temporary pain, leaving the knee loss of protection from the incomplete meniscus structure, which may accelerate

knee joint degeneration[6]. Suture or staple-based surgical repair of meniscus is only applicable in minor portion of meniscus tears, and the success rate is not satisfactory[7]. Surgical methods unable to close the injury site would result in ineffective healing of torn meniscus. Thus, a novel treatment strategy for the repair of meniscus tears is urgent.

Tissue adhesives have gained considerable attention as a promising alternative to traditional surgical sutures and staples in a variety of recent applications, owing to their ease of use, prompt response, minimal tissue damage, and the ability of adhering to irregular biological tissues[8–10]. Commercialized tissue adhesives have been widely

[1]Department of Sports Medicine of the Second Affiliated Hospital, and Liangzhu Laboratory, Zhejiang University School of Medicine, Hangzhou, China. [2]Dr. Li Dak Sum & Yip Yio Chin Center for Stem Cells and Regenerative Medicine, Zhejiang University School of Medicine, Hangzhou, China. [3]Zhejiang University-University of Edinburgh Institute, Zhejiang University School of Medicine, Haining, China. [4]China Orthopedic Regenerative Medicine Group (CORMed), Hangzhou, China. [5]Orthopedics Research Institute, Zhejiang University, Hangzhou 310009, China. [6]Sports Medical Center, the First Affiliated Hospital, School of Medicine, Zhejiang University, Hangzhou 310009, China. [7]The Fourth Affiliated Hospital, International Institutes of Medicine, Zhejiang University School of Medicine, Yiwu 322000 Zhejiang, China. [8]These authors contributed equally: Xihao Pan, Rui Li, Wenyue Li. ✉e-mail: hwoy@zju.edu.cn

accepted in the field of skin wound repair and hemostasis, which demonstrated favorable clinical efficacy[11]. However, tissue adhesives for meniscus tears repair without the assistance of sutures have not been reported yet, mainly due to their complex mechanical loading conditions and inflammatory microenvironment of the joint[12]. Differing from the tissue adhesives used for soft tissue in skin, lung, gastrointestinal tract or cardiovascular tissues, the design of adhesives for meniscus-like hard tissues necessarily integrated satisfied adhesive strength, robust mechanical properties as well as long-term regeneration function[13,14].

Silk fibroin (SF), extracted from the cocoons of silkworms, is a typical natural biomacromolecule with favorable characteristics for the formation of β-sheets, such as superior mechanical properties and good biocompatibility, making it a promising material for a wide range of fields, in the form of hydrogels and scaffolds[15]. SF also boasts abundant functional groups that can be modified through chemical reactions, and based on these, glycidyl methacrylate (GMA)-modified silk fibroin (SFMA) can be synthesized to form photocurable hydrogels and SFMA-based materials also have been developed for scarless skin repair and cartilage surface regeneration in our previous studies[16,17]. However, the adhesion and mechanical properties of SFMA-based hydrogel adhesive used in the repair of meniscus tears need to be further strengthened. Recently, ionic liquids have garnered significant attention owing to their excellent thermal stability, high electrical conductivity, and low melting point[18,19]. More importantly, the chemical structure of ionic liquids can be adjusted by changing the type of cations or anions to obtain tailor-made ionic liquids with different chemical, physical, and biological properties, offering a wide range of applications, including as biological solvents, drug delivery vehicles and for repairing skin tissue injuries[20]. Given these characteristics, incorporation with tailor-made ionic liquid capable of designing bioadhesive assemblage would be competent to meet aforementioned requirements for hard tissues aimed at meniscus tears management.

In this work, we prepare a hydrogel adhesive, S-PIL10, which involves SFMA, phenylboronic acid-ionic liquid (PIL), and growth factor TGF-β1, having the character of transitory solidification to bridge meniscus tears and expedite self-regeneration (Fig. 1). By screening variant concentrations of our own synthetic PIL, the key advance is a significant augmentation of the formation of abundant β-sheet structures and hydrogen bonds via a supramolecular polymer network, leading to the desirable mechanical properties of the adhesive product S-PIL10. The continuous release of cytokine TGF-β1 upregulates meniscus-related gene expression, and S-PIL10 itself scavenges ROS to improve pathological microenvironment for maximizing meniscal regeneration. Lastly, such integrative efficacy is prognostically assessed by an in vivo rabbit model to demonstrate an effective meniscus reconstruction without the observation of joint wear. We envision that this adhesive could be a promising product toward clinical realization, as part of meniscus tear reparative management.

## Results

### Synthesis, preparation, and characterization of S-PIL precursors and adhesives

To integrate the excellent adhesion force, anti-swelling, and tough mechanical properties into a hydrogel adhesive, a unique design and material composition was put forward in this study. SFMA was methylated from silk fibroin referred to a previous study[21], and PIL was synthesized through an alkylation reaction between 4-(bromomethyl) phenylboronic acid and 1-vinyl imidazole[22] (Fig. 2A). The structure of tailor-made PIL was capable of three features: (1) the vinyl group in the imidazole cation as a monomer copolymerized with SFMA; (2) the structure of the imidazole salt generating hydrogen bonding and promoting the formation of β-sheet structures for silk fibroin; (3) The phenylboronic acid groups reacting with hydroxy groups in the chain of silk, mainly from tyrosine and serine, to form dynamic boronic ester bonds in the network[23]. As presented in Fig. 2B and Supplementary Fig. 1A–C, the designed chemical structure of SFMA and PIL were verified by [1]H nuclear magnetic resonance ([1]H NMR) and Fourier transform infrared (FTIR). Their typical peaks were consistent with their chemical

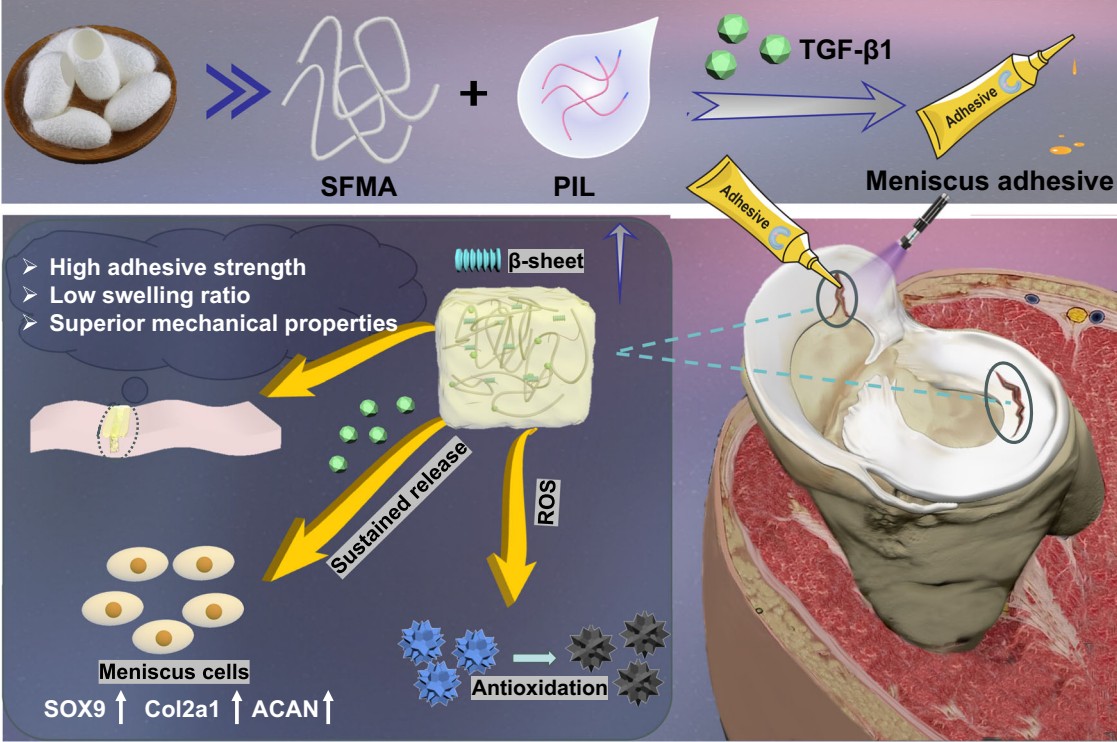

**Fig. 1 | Schematic representation of formation and application of menisucs adhesive.** The SFMA/PIL/TGF-β1 meniscus hydrogel adhesive formation process and properties of meniscus adhesive and its application in meniscus tears repair.

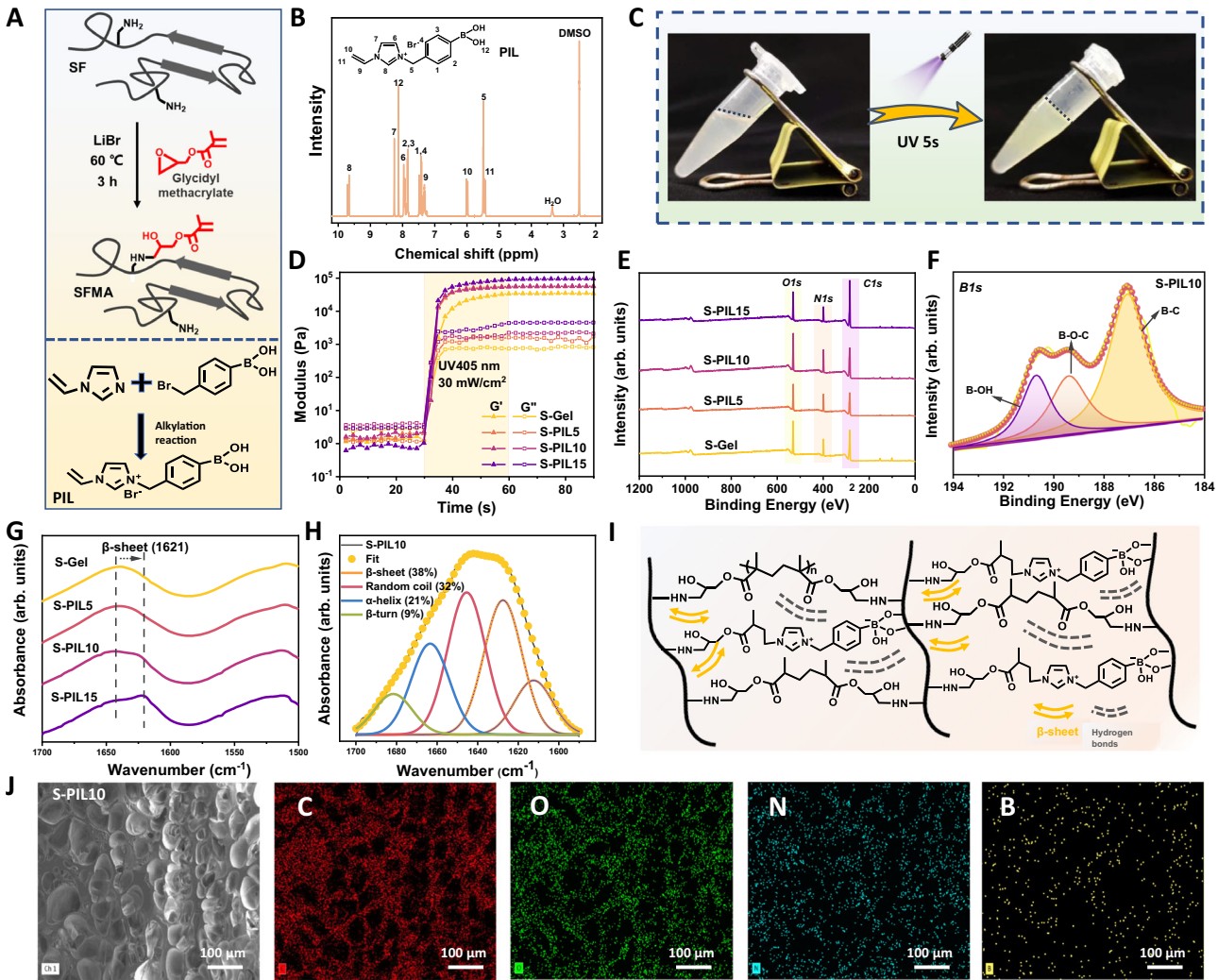

**Fig. 2 | Preparation and characterization of S-PIL precursor and hydrogel adhesives. A** Synthesis reaction of methacrylated silk fibroin (SFMA) and PIL. **B** $^1$H NMR spectra of PIL. **C** S-PIL gelation after UV light. **D** Rheology characterizations and **E** XPS full spectrum of S-Gel with the different concentrations of PIL. **F** *B1s* XPS signal of S-PIL10. **G** Amino II of S-Gel with the different concentrations of PIL in FTIR spectra. **H** Quantitative analysis of secondary structures of S-PIL10 with Gaussian curve fitting in FTIR spectra. **I** Multiple interactions between SFMA and PIL in the network. **J** SEM image and corresponding EDS elemental mapping of S-PIL10. The SEM characterization experiments were conducted three times independently.

structures and functional groups, representing δ = 6.0 ppm and 5.6 ppm in SFMA spectrum for methacrylate vinyl group signals[24]. To construct the meniscus adhesive, the contents, and proportions of these synthesized S-PIL precursors were listed in Supplementary Table 1 and these adhesives were named S-Gel, S-PIL5, S-PIL10, and S-PIL15 depending on the dosage of PIL. SFMA and PIL were then mixed and irradiated under UV light, and the meniscus adhesive solidified within 5 s through photoinitiated radical polymerization (Fig. 2C). Next verified through rheological testing and under the UV irradiation (405 nm, 30 mW/cm$^2$), the storage modulus (G′) of the adhesive increased rapidly and intersected with the loss modulus (G″), indicating instant formation of the hydrogel adhesive, which was beneficial to repair meniscus tears in-situ (Fig. 2D). Moreover, the addition of PIL did not significantly affect the gel time of the different hydrogel adhesives (Supplementary Fig. 2). Interestingly, the increasing PIL content in the hydrogel adhesives resulted in an increase in the G′ value from 35.10 kPa to 95.81 kPa, which could be attributed to the interactions between SFMA and PIL molecules and restrained movement of polymer chains in the hydrogel network[25]. To investigate these multiple interactions, the results of X-ray photoelectron spectroscopy (XPS) shown in Fig. 2E and Supplementary Table 2 showed a 2.38% increase in the

molar percentage of *B1s*, suggesting the incorporation of PIL into the SFMA hydrogel network. In high-resolution XPS spectra of S-PIL10 (Fig. 2F, Supplementary Fig. 3A, B), the new signals of B-OH, B-O-C, B-C, and C=N were observed, indicating the presence of dynamic interactions between the hydroxyl groups on SFMA and the phenylboronic acid groups of PIL[26]. Besides, FTIR results of hydrogel adhesives were presented in Fig. 2G and Supplementary Fig. 1D, and the absorption peak at 1621 cm$^{-1}$ corresponded to β-sheet structures. The secondary structure content was calculated from the amide I band, and with the addition of PIL, the molar ratio of β-sheet increased from 18% to 43% (Fig. 2H, Supplementary Fig. 4A–C). Figure 2I illustrated the anatomical networks between SFMA and PIL, including these interactions between SFMA and PIL (previously mentioned covalent bonds and abundant hydrogen bonds), and the formation of many β-sheet structures, which could be attributed to the Hoffmeister effect of PIL on silk fibroin[27,28]. Moreover, SEM images (Fig. 2J and Supplementary Fig. 5) visualized the microporous structures in these hydrogel adhesives, which could provide a favorable space for cell growth and exchange of nutrition[29]. With the increase of PIL amount, the pore size becomes smaller (from 52.53 to 32.59 μm) in Supplementary Fig. 6, which could be explained that more PIL would have more interactions with SFMA and form more

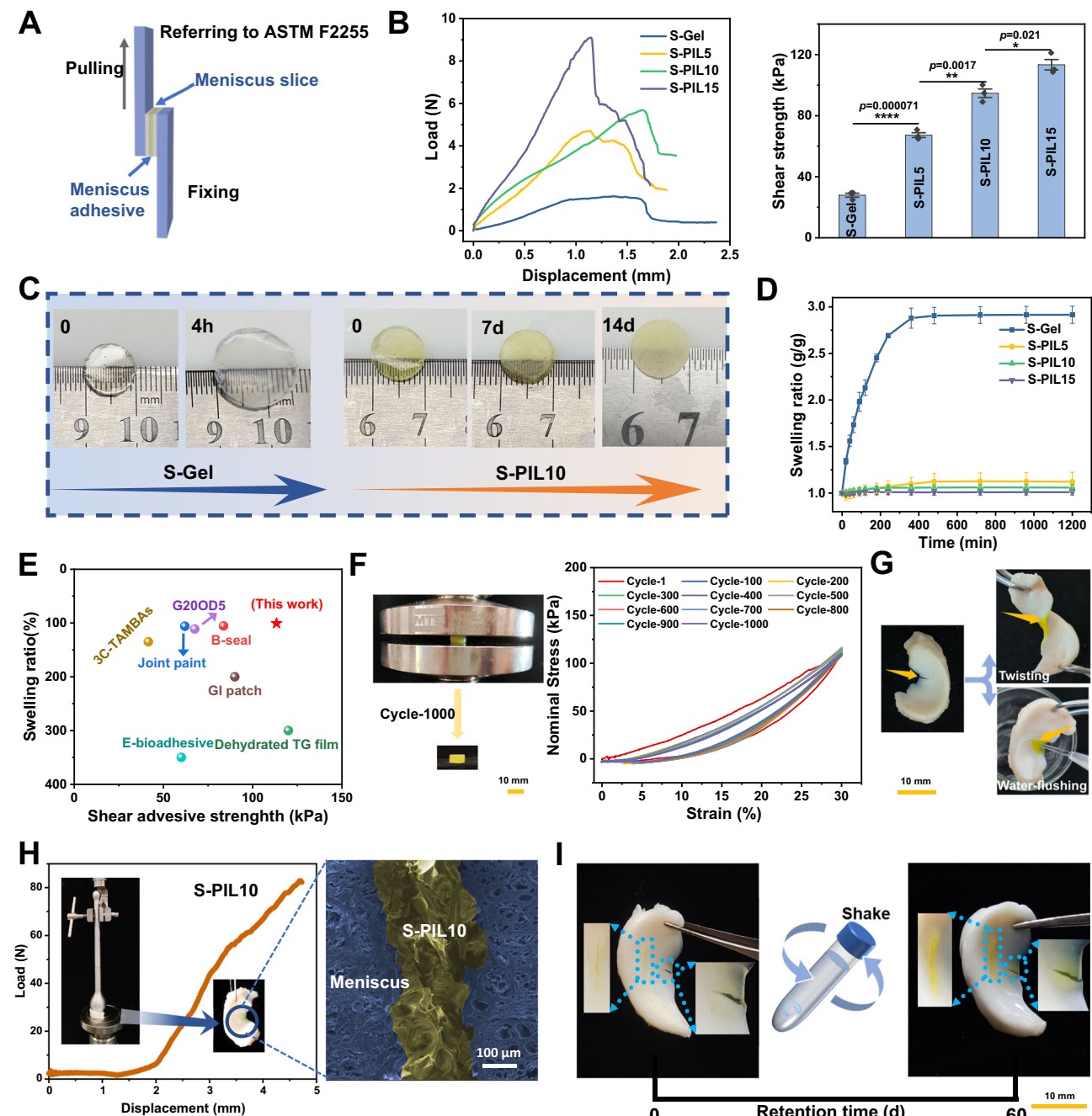

**Fig. 3 | Adhesion performance and mechanical properties of S-PIL series.**
**A** Schematic diagrams of lap shear testing on the meniscus slice **B** Load-displacement curves and lap shear strength of S-Gel with the different concentrations of PIL bonding with meniscus slice. Data are presented as mean ± SD ($n = 3$ independent experiments), and exact $p$-value was calculated with one-way ANOVA Tukey's multiple comparison test. **C** Swelling conditions and **D** swelling ratio of S-Gel and S-PIL10 in PBS buffer. Data are presented as mean ± SD ($n = 3$ independent samples). **E** Ashby plot of S-PIL10 compared with reported adhesives. **F** Macrograph and stress-strain curves of S-PIL10 with cyclic compressive loading-unloading testing for 1000 cycles. **G** Twisting and water-flushing after meniscus adhesion by S-PIL10. **H** Load-displacement curves of S-PIL gel bonding meniscus tear and SEM image of interface between S-PIL10 and meniscus. **I** Retention condition of S-PIL bonding meniscus tear after 60 days.

secondary structures and networks. Energy dispersive spectroscopy (EDS) mapping identified the presence of B element in the synthesized hydrogel adhesives, further confirming the successful construction of S-PIL hydrogel.

## Adhesion performance and mechanical properties of S-PIL series

Based on these verified multiple interactions between SFMA and our tailor-made PIL, the amplification in adhesion strength and mechanical

properties could be proposed. Hence, we moved to the experimental evaluation of the adhesion performance according to modified lap shear strength test (ASTM F2255) presented in Fig. 3A. Meniscus tissue slices were prepared and adhered through synthesized meniscus adhesive, and the resultant adhesive shear strength increased from 27.89 kPa to 113.37 kPa with the addition of PIL (Fig. 3B). Standard test methods for strength properties (ASTM F2255: Standard Test Method for Strength Properties of Tissue Adhesives in Lap Shear by Tension Loading and F2258: Standard Test Method for Strength Properties of

Tissue Adhesives in Tension) were further performed. The adhesive shear strength and tensile strength of S-PIL10 were 181.81 and 170.62 kPa (Supplementary Fig. 7A, B), respectively, which were approximately 7 and 8 times larger than the commonly used Fibrin (Supplementary Fig. 8A), and the shear adhesion of the meniscus adhesive even allowed to lift a bucket (å 5 kg) (Supplementary Movie. 1). Two other ionic liquids (1-ethyl-3-methylimidazolium bromide and 1-vinyl-3-ethylimidazolium bromide named IL1 and IL2) similarly enhanced the adhesive shear strength when added to the SFMA hydrogel (Supplementary Fig. 8B–D), however, the final adhesive ability was still lower than that of S-PIL10, highlighting the superiority of tailor-made PIL, which could be attributed to their fewer interactions with SFMA because IL1 didn't possess the vinyl group and IL2 didn't have the phenylboronic acid groups in comparison with PIL. Moreover, anti-swelling properties are crucial for ensuring tight adherence to torn meniscus tissue for an extended period of time. As shown in Fig. 3C, S-PIL10 retained its original shape and volume after 14 days, while S-Gel presented significant swelling after only 4 h. Correspondingly, the swelling ratio decreased from 2.911 (S-Gel) to 1.02 (S-PIL10) (Fig. 3D). Compared with other reported tissue adhesives[30–36], our meniscus adhesive S-PIL10 had superior shear adhesive strength and durable anti-swelling properties (Fig. 3E). These excellent properties can be attributed to the multiple interactions in the polymeric network, which enhance the mechanical properties to resist swelling. Additionally, the β-sheet structures and PIL in the network are hydrophobic, which contributes to the resistance against water absorption. Besides, studies have shown that the denser and smaller the pores of the polymeric network, the higher the anti-swelling properties of the hydrogel (Supplementary Fig. 6)[37]. Generally, other materials, such as GelMA and PEGDA, also exhibit similar anti-swelling performance due to the incorporation of PIL (Supplementary Figs. 9 and 10). Notably, the anti-swelling properties of HAMA hydrogel were significantly improved, further verifying the superiority of tailor-made PIL. Then the stress-strain curves of fabricated hydrogel adhesives about compression performance supported the maximum stress and compressive modulus increased along with the introduction of PIL (Supplementary Fig. 11). To evaluate functions of the adhesive in biophysical environment of the meniscus, a 1000-cycle loading test was performed and the hydrogel didn't show an obvious change in shape or mechanical strength (Fig. 3F), demonstrating its robust resistance to fatigue and stability of physical strength[38]. And with the increasing compression cycles, S-PIL10 exhibited a decline in energy dissipation (Supplementary Fig. 12), which was similar to meniscus and its derived materials. When the torn meniscus was adhered by S-PIL10, it could lift a bottle more than 1 kg (Supplementary Fig. 13) and keep robust adhesion under twisting and water-flushing conditions (Fig. 3G). We also exposed the adhered meniscus to biomechanical force to mimic the loading conditions within the knee, and showed no significant changes (Supplementary Movie. 2). The adhesive was able to withstand biomechanical force up to 80 N, meeting the experimental demands of New Zealand white rabbits (Fig. 3H). SEM images showed that S-PIL10 integrated well with the meniscus tissue, demonstrating mechanical interlocking effect between photocured hydrogel adhesives and meniscus. Furthermore, adhesion of S-PIL10 to meniscus with radial or longitudinal tears was still steady after 60 days (Fig. 3I). Based on above excellent performance, the instant robust hydrogel adhesive could be used for the subsequent repair of meniscus tears.

### Biocompatibility and ex vivo therapeutics of meniscus adhesives

The biocompatibility was evaluated to ensure the safety of the meniscus adhesives. As presented in Fig.4 A–C, rabbit meniscus cells maintained good cell viability and apparent proliferation co-cultured with S-PIL series. On Day 7, each well in the 24-well plate was overspread with rabbit meniscus cells. Similarly, L929 fibroblasts also

exhibited high cell viability (>95%) on the surface of S-PIL series (Supplementary Fig. 14). However, S-PIL15 had a slight inhibition on the proliferation of rabbit meniscus cells, as PIL itself may be not very friendly to cells, and with the increase of the concentration, S-PIL series would have negative effect on the cells. PIL was added into the hydrogel adhesive and had some interactions in the polymeric network to prevent the leakage of PIL to have good biocompatibility. Integrating the above results, S-PIL10 was considered as the balanced formulation with superior mechanics and cytocompatibility, which was optimally selected in the following meniscus adhesive study. Further, in vivo biocompatibility and degradation test of meniscus adhesives was performed subcutaneously (Supplementary Fig. 15A). Silk fibroin is a well-known biomaterial with low immunogenicity and there was almost no significant difference in inflammatory response among S-Gel and S-PIL10 groups. Inflammatory cells were observed around the implants in the first week and gradually disappeared over time (Supplementary Fig. 15B), the acute inflammation disappeared and the chronic inflammation was almost invisible, suggesting that the hydrogel adhesive and its degradation product have good biocompatibility. And the morphology of S-PIL10 showed mild degradation, but the weight of S-Gel changed significantly within 8 weeks (Supplementary Fig. 15C). S-Gel first experienced significant swelling with the maximal size achieved at day 3 and then shrank in 8 weeks, and the mass of S-PIL10 in vivo increased slightly in the beginning, possibly caused by the rising of water content, consistent with previous swelling results. Subsequently, the mass of S-Gel and S-PIL10 was reduced, probably due to a slight degradation of the hydrogel adhesives. During the degradation process of S-Gel and S-PIL10, the swelling behavior of hydrogels could not be ignored. Compared with S-PIL10, S-Gel has more body fluid in the polymeric network, which contributes to the higher mass remaining of S-Gel, and the mass remaining of S-PIL10 was lighter with less water due to its excellent anti-swelling properties. These factors jointly resulted in the observed differences in degradation process between the two hydrogels. Notably, the structure of S-PIL10 at week 2 was cracked and significant cells proliferated between inter-hydrogel space, indicating the ideal biocompatibility and repair potentials of S-PIL10 (Supplementary Fig. 15C).

The excellent mechanical properties and biocompatibilities kept adhered S-PIL10 stable on torn meniscus, in the meanwhile, growth factors or cytokines were formulated into the hydrogel to further promote tissue self-regeneration. Multiple growth factors selected from meniscus regeneration-related studies were screened according to their promoting effects on meniscus cells (Supplementary Fig. 16). Alcian blue staining and immunofluorescence staining proved transforming growth factor-beta 1 (TGF-β1) and TGF-β3 owned significant promoting effects on the cell proliferation and collage secretion of meniscus cells. TGF-β3 has widely been reported to delay the degeneration of meniscus in previous studies[39,40], while the function of TGF-β1 on meniscus was still unknown that interested us profoundly[41]. For this purpose, TGF-β1 was included into S-PIL10 composition to fabricate the meniscus adhesive. As presented in Fig. 4D, E, TGF-β1 loaded S-PIL10 significantly enhanced the expression of meniscus-related genes in meniscus cells compared with neat S-PIL10 group, like *SOX9*, *Col2a1*, and *ACAN*. A long-term healing process of meniscus tears was common, so TGF-β1 continuously released for months was advantageous. The release curve of TGF-β1 loaded S-Gel and S-PIL10 demonstrated sustained release in more than 8 weeks (Fig. 4F) due to the noncovalent interactions between TGF-β1 and these polymeric networks. Also, TGF-β1 was released more slowly in S-PIL10 compared with the release performance in S-Gel due to the addition of PIL, which could be explained by anti-swelling ability of S-PIL10 and the smaller size of pores compared with S-Gel. These results were consistent with the above-mentioned physicochemical characterizations, implying that TGF-β1 loaded S-PIL10 was beneficial for in vivo repair and prognosis. Inflammation

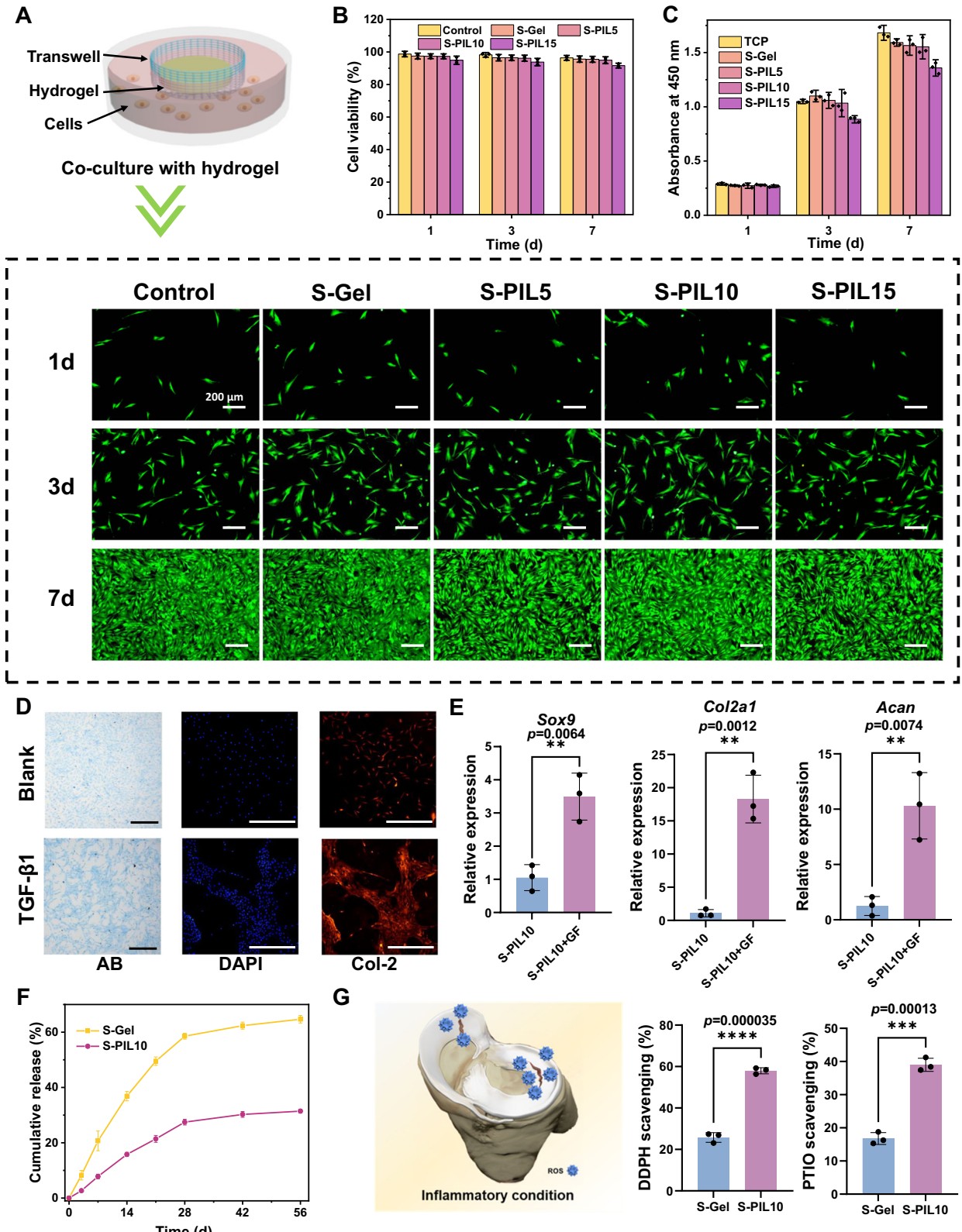

**Fig. 4 | Biocompatibility and ex vivo therapeutics of meniscus adhesives.**
**A** Schematic illustration of co-culture with hydrogels and live/dead cell assays.
**B** Cell viability of rabbit meniscus cells co-cultured with hydrogels within 7 days.
Data are presented as mean ± SD ($n = 3$ independent cell experiments). **C** Cell
Counting Kit 8 (CCK-8) assays of rabbit meniscus cells co-cultured with hydrogels
within 7 days. Data are presented as mean ± SD ($n = 3$ independent cell experi-
ments). **D** Alcian blue (AB) staining and immunofluorescence staining of meniscus
cells. **E** Meniscus-related gene (*Sox9, Col2a1,* and *ACAN*) expression in the S-Gel and

S-PIL10 at 2 weeks. Data are presented as mean ± SD ($n = 3$ independent cell
experiments) and exact *p*-value was calculated with two-tailed student's *t*-tests.
**F** TFG-β1 release-time curves of S-Gel and S-PIL10. Data are presented as mean ± SD
($n = 3$ independent samples). **G** Inflammatory condition after meniscus tears and
DDPH scavenging and PTIO scavenging of S-Gel and S-PIL10. Data are presented as
mean ± SD ($n = 3$ independent hydrogels) and exact *p*-value was calculated with
two-tailed student's *t*-tests. Scale bars are 200 μm.

is a natural response to meniscus tears when not untreated, that will impede the repair process. The designed meniscus adhesive ideally has antioxidant capacity to achieve inflammation control[42,43]. The antioxidant activity was quantitated by the 1,1-diphenyl-2-picrylhydrazyl (DPPH) and 2-Phenyl-4,4,5,5-tetraMethyliMidazoline-3-oxide-1-oxyl (PITO) assay. As presented in Fig. 4G, S-PIL10 performed better scavenging ability against different free radicals compared with S-Gel, which could be attributed to the dynamic borate ester bonds of tailor-made PIL and SFMA[44]. To further confirm the antioxidant activity of hydrogel at the cellular level, the cell protection efficiency of prepared hydrogels by clearing ROS was evaluated through the DCFH-DA kit. As shown in Supplementary Figs. 17 and 18, the S-PIL10 group exhibited the weakest green fluorescence and their lowest intracellular ROS levels, suggesting that S-PIL10 had prominent ROS scavenging activity and could protect the cells from being oxidatively damaged in vitro. Collectively, TGF-β1 loaded S-PIL10 exhibited comprehensive biological functions for meniscus repair.

## The application of S-PIL10 on in vivo multiple types of meniscus tears

Among various types of meniscus tears, radial tear in white–white zone is the most challenging one due to the lack of blood flow in the region[4,45]. The fabricated S-PIL10 was typically applied to radial tear and the entire experimental procedures including creating meniscus tear model and in-situ adhesive operation were shown in Fig. 5A. The tear model of the medial meniscus was penetrating and the control group received no treatment. After 8 weeks, the meniscus was collected for observation and histological examination. As presented in Fig. 5B, macroscopic views of the rabbit meniscus showed no significant tear in the S-PIL10+GF group but apparent fissure could still be visualized in other experimental groups. H&E staining and immunofluorescence staining of the meniscus presented the same results (Fig. 5C). In the control group, meniscus tears were unable to be self-healed, and significant large tears remained. The S-PIL10+GF group showed successful repair of meniscus radial tears, while the group

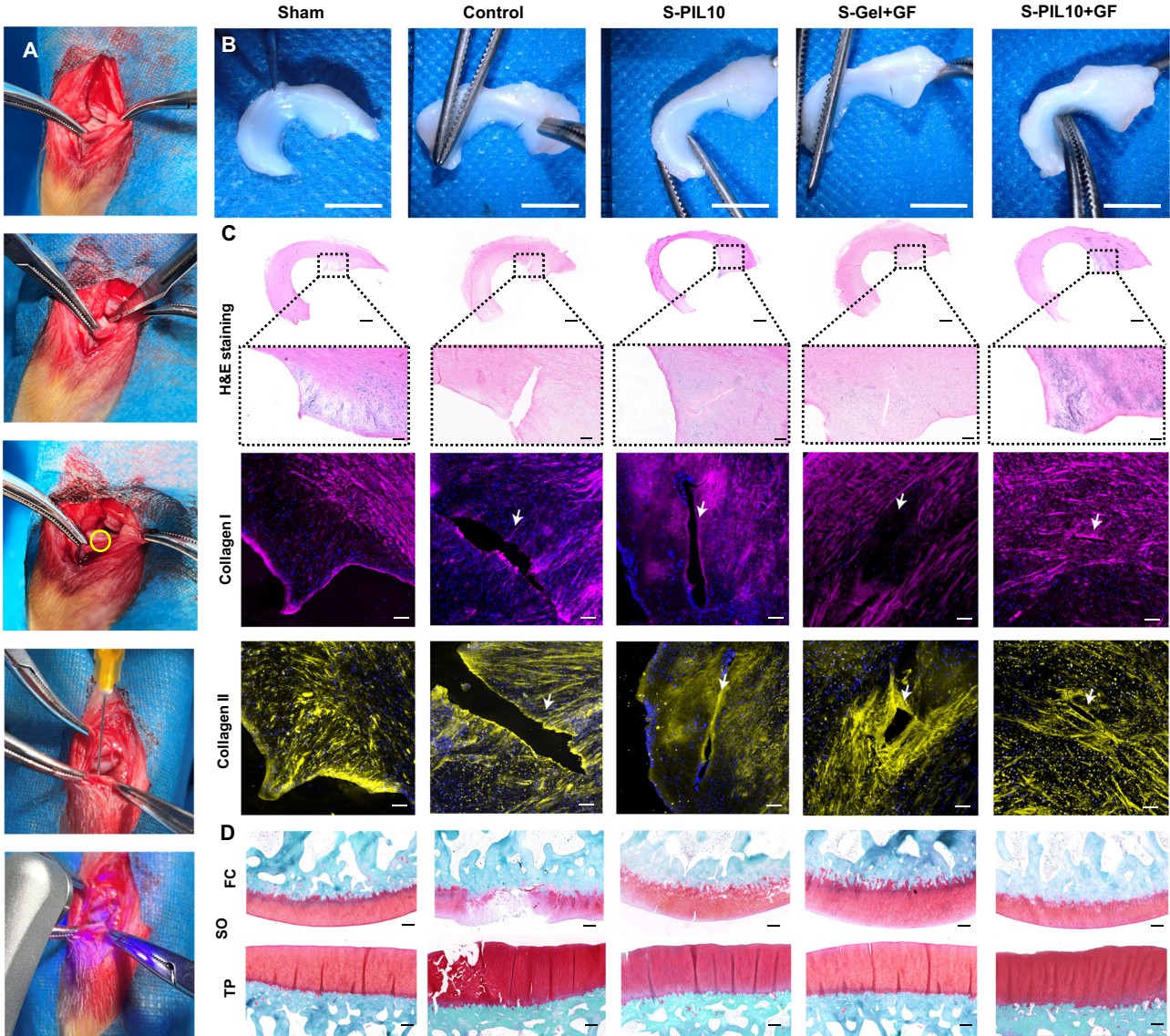

**Fig. 5 | Repair effect of meniscus tears and evaluation of in vivo articular cartilage wear after 8 weeks in vivo. A** Process of meniscus tears modeling and adhesives repair: step 1: transection of the medial collateral ligament; step 2: exposure of meniscus toward the femur; step 3: incision of the meniscus through the full thickness; step 4. In-situ sealing by S-PIL adhesives; step 5: solidification by UV and wound closure. **B** Macrograph of meniscus after meniscus radial tears of two months **C** Hematoxylin & eosin (H&E) staining and immunofluorescence on collagen I and collagen II of repaired meniscus with radial tears. **D** Safranin O/fast green staining (SO) of femoral condyles (FCs) and the tibial plateaus (TPs) after meniscus radial tears. For **C, D** the animal experiment was repeated three times independently. Scale bars in **B** are 5 mm, in the first row of **C** are 1 mm, and others are 200 μm.

using S-PIL10 and S-Gel+GF both showed partial healing with smaller tear size, which proved the superiority of our designed meniscus adhesive. Besides, S-PIL10+GF group had the smallest area of tear based on statistical results of histological staining (Supplementary Fig. 19A). These results could be explained as followed: (1) the strong adhesion performance of S-PIL10 can adhere the meniscus tears tightly, and the insufficient adhesion strength of S-Gel itself couldn't be maintained for a long time, which verified the importance of mechanical properties of meniscus adhesive; (2) addition of TGF-β1 can promote the regeneration of meniscus tears in white–white zone, which is also crucial for meniscus repair.

The purpose of meniscus tears repair is known to prevent cartilage degeneration and osteoarthritis, thus examination of articular cartilage is important in the repair process[46]. As shown in Fig. 5D and Supplementary Fig. 19B, meniscus tears led to local wear and tears at the site of femoral condyles (FCs) and the tibial plateaus (TPs) in the control group after 8 weeks, while there is little sign of osteochondral damage on the FCs and TPs of the S-PIL10+GF group with the lowest Osteoarthritis Research Society International (OARSI) score (Supplementary Fig. 19C), indicating the prominent osteochondral protective effects[47]. Lastly, the longitudinal tear is another common type of meniscus tear[48]. Again, S-PIL10+GF showed the repeatable excellent performance in the longitudinal tear model, indicating its suitability for various meniscus tears (Supplementary Fig. 20). Taken together, S-PIL10 meniscus adhesive can markedly repair meniscus tears and protect cartilage against wearing.

## Discussion

Meniscus plays unique roles in the knee joint enabling load transmission, stability, and lubrication[49]. Unfortunately, meniscus tears happen commonly in daily life and the repairing is challenging due to the stress-loading environment and its heterogeneous distribution of composition, particularly the absence of blood vessels in (white–white zone)[50]. The key purpose of this work lay in the revolutionary strategy of meniscus tear therapy and was to verify whether the concept of meniscus adhesive was feasible and provide a novel clinical method that could repair various types of meniscus tears due to inefficacy in current treatments. For example, partial or total meniscectomy and defect closure with sutures have non-negligible drawbacks that limit their success rate of application, and the former would reduce the force area to bring more load on the cartilage while the latter fails to repair the meniscus in the white–white zone from the point of regeneration[1]. Moreover, the reported meniscal scaffolds like the Menaflex Collagen Meniscus Implant and Actifit scaffolds were used to substitute the torn meniscus and showed unsatisfactory outcomes, especially without observing their long-term effectiveness[51]. Recently, Wang and his group have used the pulsed electromagnetic field to enhance healing of a meniscus tear in a small-animal model, without validating the repeatable therapeutics in medium or large-animal models[52]. Thus, tissue adhesives indicated a highly clinical therapeutic method to repair meniscus tears, as the strategy saved the meniscus to preserve the integrity of the native meniscus tissue, and ensured tight adhesion of the torn meniscus tissue until complete healing. In 1995, cyanoacrylate (the main component of 502 glue) based adhesive was first used for the repair of meniscal tears. Though in vitro experiments showed good performance, this glue cannot be used for meniscal repair in vivo due to its cytotoxicity and severe inflammatory reactions[53]. Then, fibrin, a natural polymer material, was used in the repair of meniscal tears. However, fibrin had weaker mechanical properties, and lower adhesive capacity, resulting in poorer repair of meniscal tears[54]. Furthermore, block copolymer bioadhesives based on trimethyl vinyl (TMC) were also used to repair meniscal tears. This hydrogel adhesive had stronger strength than fibrin glue, and the elastic modulus also reached the level of natural meniscus[55], but biocompatibility still needs to be solved. Actually, these reported

adhesives were used to adhere meniscus tears in vitro, and few cases using hydrogel adhesives for the successful repair of meniscus tears in vivo without the assistance of sutures have been reported due to the repair difficulties and the demand for the mechanics[7,56]. In the current work, we aimed to develop an instant robust hydrogel adhesive based on SFMA, PIL and TGF-β1 to enable sealed-tight reconstruction of meniscus tears.

In the first part, we focused on enhancing the mechanical properties of the meniscus adhesives S-PIL10. Among these properties, the adhesive properties are the most significant in the meniscus repair and improved mainly from two perspectives. On the one hand, an in-situ photocurable instant S-PIL10 was fabricated, which fulfilled the gaps between the torn meniscus tissue and formed the mechanical interlocking to improve interfacial adhesion[57]. Actually, noncovalent interactions play an important role in achieving fast interfacial bonding and contributing to adhesion stability[58]. S-PIL10 could provide noncovalent interactions in tissue adhesion including electrostatic interactions and hydrogen bonds due to its various active functional groups. On the other hand, the hydrogel adhesive itself was strengthened through selecting robust silk fibroin and designing tailor-made ionic liquid[59]. The unique ionic liquid reacted with SFMA by multiple interactions, including copolymerization with the vinyl group, production of dynamic boronic ester bonds through the phenylboronic acid groups reacting with hydroxy groups, and the generation of hydrogen bonding and formation of many β-sheet structures through the Hoffmeister effect of PIL on silk fibroin[60]. These interactions significantly enhance the adhesive properties of S-PIL10. Compared with other tissue adhesives reported in top journals, our meniscus adhesive S-PIL10 demonstrated superior and competitive adhesion properties[32,34,61]. Besides, this instant robust silk fibroin hydrogel adhesive had anti-swelling properties and enabled good long-term adhesion, which satisfied the mechanical requirements of rabbit meniscus tears, making it suitable for the repair of meniscus tears.

In the second part, apart from the excellent mechanical properties, the qualified hydrogel adhesive should promote the repair and regeneration of the adhered torn meniscus in consideration of the actual in vivo condition due to the self-unhealable characteristic in white–white zone. As reported, Wei designed a hydrogel-elastomer hybrid as adhesive patch to promote the meniscus repair in red–red zone, but the repair in white–white zone was mainly with the aid of growth factors or cytokines[39,45]. Hence, the growth factor TGF-β1 was loaded into the meniscus adhesive S-PIL10 to promote effective healing in the avascular region. S-PIL10 constantly released the factor and upregulated meniscus-related gene expression like *SOX9*, *Col2a1,* and *ACAN*. Moreover, inflammation is common to meniscus tears in actuality, but it is ignorable in the repair of meniscus which affects the repair results[48]. The hydrogel adhesive S-PIL10 was designed to possess anti-inflammatory properties by incorporating dynamic boronic ester bonds of tailor-made ionic liquid and hydroxy groups, which cleared reactive oxygen species (ROS) and improved the microenvironment of the torn meniscus[62]. Equipped with these superior properties, the hydrogel adhesive S-PIL10 presented excellent repairing performance in various types of meniscus tears of New Zealand white rabbits and demonstrated that the strong adhesion properties of S-PIL10 and the function of TGF-β1 complemented each other in the repair process. However, there were still several limitations in our research. The meniscus of rabbit model was different from the human meniscus, and the larger animals should be used in future studies and the repairing performance could be more convincing in clinical treatment. Besides, the rabbits were treated with the meniscus adhesive S-PIL10 immediately after surgery, while the chronic meniscal lesions were more commonly seen in the clinic[52], and the therapeutic effect of S-PIL10 on inconspicuous and gradual tears needed to be further verified.

In summary, a novel meniscus adhesive S-PIL10 has been developed through the combination of SFMA, PIL, and TGF-β1. This adhesive possesses instant gelation, superior mechanical strength, strong adhesion, anti-swelling properties, and fatigue resistance due to the addition of tailor-made PIL. Besides, this adhesive induces the upregulations of *SOX9*, *Col2a1*, and *ACAN* genes, and renders the microenvironment beneficial for healing by scavenging ROS. The meniscus adhesive successfully repairs meniscus tears without causing joint damage in vivo. Overall, this meniscus adhesive proves itself a promising material and revolutionary clinical method for in-situ meniscus tears repair.

## Methods

### Ethical regulations statement

All animal experiments have been approved by the Animal Experiment Committee of Zhejiang University and fully comply with the National Institutes of Health Guide for the Care and Use of Laboratory Animals (AIRB-2022-0483).

### Synthesis and characterization of SFMA

50 g of raw silk (Bombyx mori) was degummed by boiling in 2 L of 0.05 M sodium carbonate solution for 45 min and then washed clearly. After degummed silk was dried at ambient temperature, the 30 g of above silk was dissolved in 150 mL of 9.3 M lithium bromide solution at 60 °C for 1 h. Then, 30 mL glycidyl methacrylate solution was added to the mixture at a rate of 0.5 mL/min. Subsequently, the resulting solution was filtered through suction filtration and dialyzed against distilled water using 8–14 cutoff dialysis membranes for 4 days. After the solution was frozen for 4 h, SFMA foam was formed by lyophilization for 4 days (stored at −80 °C until further use).

500 μg of SFMA foam was dissolved in 500 μL of deuterium oxide ($D_2O$, Sigma–Aldrich). Proton nuclear magnetic resonance ($^1H$ NMR) at a frequency of 400 MHz with a Spectrometer (NMR, AVANCE NEO 400, Bruker, Germany) was used to determine successful modification of SFMA.

Fourier transform infrared spectroscopy (FTIR, Vertex 70, Bruker, Germany) was used to record samples of FTIR spectra (500–4000 $cm^{-1}$) with 32 wavelength scans at a resolution of 4.0 $cm^{-1}$. After subtracting the baseline from amide II (1600–1700 $cm^{-1}$), the amide II region of SFMA was analyzed by the second derivatization and further Gaussian curve fitting via PeakFit v4.12 software. The wavenumber of individual component peaks indicated different secondary structures.

### Synthesis and characterization of tailor-made ionic liquid

Firstly, 0.82 g of 4-(bromomethyl)phenylboronic acid and 1.41 g of 1-vinyl imidazole were mixed in 30 mL of acetonitrile. The mixture reacts in a nitrogen environment at a temperature of 55 °C for 12 h. Subsequently, excess solvent was removed by rotary evaporation and the resulting mixture was washed with a large amount of ether. The purpose of this washing step was to achieve a pH value of 10. The obtained ionic liquids were stored at −4 °C for future utilization. The $^1H$ NMR (NMR, AVANCE NEO 400, Bruker, Germany) and FTIR spectroscopy (FTIR, Vertex 70, Bruker, Germany) were used to confirm the successful synthesis of ionic liquid.

### Fabrication and characterization of meniscus adhesives

The meniscus adhesives precursor solution containing 30% (w/w) SFMA, 0.5/1/1.5% (w/w) and 0.25% (w/w) LAP. The resulting solution is then exposed to UV irradiation to form a hydrogel adhesive (S-Gel, S-PIL5, S-PIL10, and S-PIL15), respectively.

The chemical composition and microstructure of these adhesives were characterized using XPS spectroscopy (XPS, K-Alpha, Thermo Fisher Scientific, USA), FTIR (FTIR, Vertex 70, Bruker, Germany), SEM, and EDS element mapping (M-30+, COXEM, South Korea). In rheological studies, time-sweep oscillatory tests were performed at a frequency of 10 Hz, a strain of 3% and a gap of 500 μm. The storage modulus (G′) and loss modulus (G″) of the samples were recorded with 30 mW/cm² UV light (405 nm) between 30 s and 60 s.

Chemical compositions were analyzed by the X-ray photoelectron spectroscopy (XPS, K-Alpha, Thermo Fisher Scientific, USA) and the FTIR (Vertex 70, Bruker, Germany) within the range of 500–4000 $cm^{-1}$ with 32 wavelength scans at a resolution of 4.0 $cm^{-1}$.

Before scanning electron microscope (SEM), the hydrogels were frozen at −30 °C and then freeze-dried. The samples were cut off to expose the cross-section. The samples were sprayed with gold for three minutes. The cross-sectional morphology of the samples was observed through SEM (EM-30+, COXEM, South Korea). The average diameter of pores was calculated using ImageJ software. To evaluate the element distribution in the hydrogel, EDS element mapping was performed in the suitable cross-section. The scalpel was used to cut the meniscus of pigs to model meniscus tears. Then meniscus adhesive was glued meniscus tear. The SEM was used to observe the microstructure of the interface between adhesive and meniscus. The ROS scavenging activities of meniscus adhesives were measured via DPPH and PITO standard operating procedures (NanoDrop instrument, Thermo Fisher Scientific, USA).

### Mechanical test

The mechanical properties of the samples were tested via a mechanical tester (Instron 5943, Instron, USA). To show the adhesive properties, lap shear test was performed referring to ASTM F2255. The casing or meniscus slice was adhered to the glass slide. 25 μL of precursor solution was spread on the tested substrate, and then UV irradiation for 30 s formed the hydrogel adhesive. The test specimens were placed in the grips with cross-head speed of 5 mm/min until their breakage. The maximum values of load were recorded. The bonding area of meniscus is 0.6175 $cm^2$ and that of casing is 2.5 $cm^2$. Three samples were measured to calculate the lap shear strength of each group.

$$\text{Lap shear strength} = F/A$$

where F is the maximum load and A is the bonding area.

The $10 \times 10 \times 2\ mm^3$ cube-shaped specimens were made as test hydrogels. The compressive rate was 2 mm/min and the strain level was up to 95% strain. The stress-strain curves were recorded using Instron 5943 with a 1 kN sensor. The compression modulus was calculated as the slope of the stress-strain curve between 10% and 20%. The cyclic compression load-unload test was performed with the strain of 30% for 1000 cycles without resting. A metal grinding pestle with appropriate size was used to simulate the pressure of the femur on the meniscus and test the adhesion performance of the adhesive to the meniscus.

### Swelling properties

To determine the swelling properties, the above hydrogel samples were immersed in PBS solution at 37 °C. The initial mass ($W_d$) of the hydrogels was recorded before immersion, and the mass after immersing at each time point ($W_s$) was measured. The swelling rate was calculated through the following formula.

$$\text{Swelling index}(\%) = \frac{W_s - W_d}{W_d}$$

Meniscus adhesive was used to adhere circumferential tear and radial tear of the meniscus. To characterize the long-term underwater adhesion of meniscus adhesive, the adhered meniscus was immersed in PBS solution at 37 °C for 60 days.

## Animal experiments

About 200–220 g of male Sprague-Dawley rats were subjected to general anesthesia with tribromoethanol. The back of the rat was shaved and wiped with 75% alcohol. Four 1 cm incisions on the back were made, and each sample was placed in a separate subcutaneous pocket. Four samples were taken at each time point (3d, 1w, 2w, 4w, and 8w), three samples were weighed, and one sample was used for H&E staining.

The joints of the male New Zealand White rabbits (aged 16 weeks old) were shaved and wiped with 75% alcohol. The skin tissue and fascia tissue were cut to create a 2 cm incision. Then, the medial collateral ligament was cut off and the joint cavity was open. The scalpel was used to penetrate the anterior horn of the medial meniscus. After modeling, the precursor solution was injected into the wound of the meniscus via a 1 ml syringe, and then formed a hydrogel immediately through the ultraviolet light to bond the meniscus tears. After 8 weeks, the rabbits were euthanized, and their joints were removed for further evaluation.

## Histological analysis and assessment

The rabbit meniscus and joints were fixed in excessive 4% (w/v) paraformaldehyde for one week. Then rabbit joints were decalcified by 10% (w/v) ethylenediaminetetraacetic acid disodium salt (EDTA-2Na) solution for three months and meanwhile, EDTA-2Na was renewed every week. The rabbit meniscus and joints were continuously sectioned with a thickness of 7 μm via microtome cryostat. The H&E, Safranin O/fast green staining, and immunofluorescence were performed based on the standard protocols. Histological images were collected by Pannoramic MIDI scanner (3D HISTECH, Hungary). The femoral condyles (FCs) and the tibial plateaus (TPs) were blindly and independently evaluated by three researchers according to the Osteoarthritis Research Society International (OARSI) osteoarthritis cartilage histopathology assessment system, in which a higher score suggested worse cartilage condition.

## Statistical analysis

All experiments were conducted at least three times independently, and the results were expressed as mean ± SD. All statistical analysis was performed using Origin 2023 software and GraphPad Prism9.0. The significance between two groups was analyzed using two-tailed student's $t$-tests. Multiple comparisons were performed using Tukey's post hoc test and one-way analysis of variance (ANOVA). For all tests, $*p < 0.05$, $**p < 0.01$, $***p < 0.001$, and $****p < 0.0001$ indicate statistical significance, ns: no significance.

## Reporting summary

Further information on research design is available in the Nature Portfolio Reporting Summary linked to this article.

## Data availability

All relevant data are available within the article and Supplementary information. Any additional requests for information can be directed to and will be fulfilled by the corresponding authors. Source data are provided with this paper.

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

## Acknowledgements

This work was supported by the National Key Research and Development Program of China (2023YFE0206700 to H.O.), and the National Natural Science Foundation of China (T2121004 to H.O., 92268203 to H.O.), and the China National Postdoctoral Program for Innovative Talents (BX20220266 to X.P.). The authors sincerely thank Dr. Wei Wei for the valuable advice in improving the article.

## Author contributions

X.P. and H.O. conceived and designed the project. X.P. and R.L. synthesized and characterized the adhesives. X.X. and C.X. assisted in adhesive characterization. W.L., W.S., Y.Y., J.F., Y.L., and Y.C. performed all the in vitro and in vivo experiments. X.P., R.L., and W.L. discussed the results, interpreted the data, and wrote and revised the paper. X.W., X.Y., and H.O. contributed to the manuscript revision and data analysis. All authors read and approved the final version of the manuscript.

## Competing interests

The authors declare no competing interests.
