## [Peer Review File · Nature Communications]

Reviewers' Comments:

Reviewer #1:

Remarks to the Author:

The study presented in this paper is interesting, well carried out and well presented. The conclusions are based on evidences presented herein. This work is without any doubt worth for publication in some journal; the originality is limited to the use of silk fibroin, modified with a well-known procedure (vinyl imidazole) and the tissue engineering is based on well known growth factor tgf-beta.

Reviewer #2:

Remarks to the Author:

In this manuscript, Pan et al. present a novel hydrogel adhesive utilizing silk fibroin and phenylboronic acid-ionic liquid (PIL). The material presented here is quite intriguing as it combines the components of a radical scavenger, tissue adhesive, and drug delivery carrier, which are very important in meniscal tear repair and regeneration. The authors have presented a detailed characterization of the synthesized material and the biocompatibility data presented here is also appropriate. Overall, the manuscript is well organized with detailed discussion. I do have the following comments and questions which should be addressed to improve the manuscript.

Major concerns:

1. The tissue adhesive property is essential for the material in this work. The authors have compared the adhesive results with commercial products and lab-made other types of IL. However, why was this PIL chosen to enhance the properties of the adhesive and the adhesion mechanism should be discussed in detail.
2. The meniscus adhesive solidified within 5s and the time is short. Does the addition of PIL affect the gelation time of SFMA hydrogel?
3. It is interesting the hydrogel has anti-swelling properties when using PIL in this work. Can the authors comment on the mechanism of this phenomenon? Can this PIL be used for other materials such as GelMA or PEGDA and lead to the same anti-swelling property?
4. S-PIL10 has good compression performance. However, the energy dissipation is also important for meniscus. The energy dissipation of S-PIL10 needs to be further demonstrated in comparison with S-Gel during loading-unloading tests for 1000 cycles.
5. The authors demonstrated the antioxidant activity of the material using DPPH and PITO assay. How about the antioxidant activity in vitro using cells?
6. In histology results, the selection of cartilage positions in the tibia seems inconsistent, and the same location should be chosen for comparison.

Minor comments:

1. In section Adhesion performance, the authors stated they used a standard test method. Strictly, modified methods were used and should be revised because there are differences between the authors' methods and standard methods.
2. "As shown in Fig. 2C, S-PIL10 retained original shape and volume after 14 days, while S-Gel presented significant swelling after only 4 hours, and the swelling ratio decreased from 2.911 to 1.02.". This statement is confusing and should be clearer.
3. Please rewrite "Differing from soft tissue in skin, lung, gastrointestinal tract or cardiovascular tissues, the design of adhesives for meniscus-like hard tissues necessarily integrated satisfied adhesive strength, robust mechanical properties as well as long-term regeneration function" in the introduction section, it is confusing.

4. In the section 'Biocompatibility and ex vivo therapeutics of meniscus adhesives', Did the authors perform studies directly seeded cells on the surface of the adhesives or within the adhesives? Why did S-PIL15 have a slight inhibition on the proliferation of rabbit meniscus cells? Does it concentration dependent?

5. In the section 'Biocompatibility and ex vivo therapeutics of meniscus adhesives', Is there any difference in inflammatory response among groups? I cannot see from the low mag image. How about the chronic inflammation?

6. In the section 'Biocompatibility and ex vivo therapeutics of meniscus adhesives', "... , probably due to a slight degradation of the hydrogel adhesives. Notably, the structure of S-PIL10 at week 2 was cracked and significant cells proliferated between inter-hydrogel space, indicating the ideal biocompatibility and repair potentials of S-PIL10 (Fig. S10C)...". From the macroscopic images, there is slight shrinkage in adhesive size by 10 weeks, indicating that these adhesives might have relatively slow degradation rates. What is the optimal degradation of adhesive for this application? Did the authors perform any degradation studies?

7. In the section 'Biocompatibility and ex vivo therapeutics of meniscus adhesives', "Also, TGF- β 1 was released more rapidly in S-Gel compared with the release performance in S-PIL10 due to the addition of PIL". It is better to state that S-PIL had slow release. Does TGF-1 crosslinked with other components during the UV treatment?

8. In the section 'The application of S-PIL10 on in vivo multiple types of meniscus tears', the authors stated that "Besides, S-PIL10+GF group had the smallest area of tear based on statistical results of histological staining (Fig. S12A)". Given the smallest area of tear, do the authors expect whether it will completely repair the tear in a longer time or not?

9. Some of the methods are not completed and should be added in Supplementary Information.

Reviewer #3:

Remarks to the Author:

Silk Fibroin Hydrogel Adhesive Enables Sealed-tight 2 Reconstruction of Meniscus Tears

NCOMMS-23-47640

In this manuscript by Pan et al., the authors have designed a material system based on silk fibroin that is composed of methacrylated silk crosslinked with phenylboronic acid-ionic liquid loading with growth factor TGF- β 1, which exhibits instant gelation, demonstrates strong adhesion, swelling resistance, and anti-fatigue capabilities, compared to neat silk fibroin gel. Further, the authors used them to repair the meniscus as meniscus adhesive which facilitates healing by scavenging reactive oxygen species and upregulation of different genes. The material system is not completely novel as methacrylation of silk is widely reported (Nature Protocol, 2021, 16, 5484, ACS Biomater Sci. Eng., 2019, 5, 12, 6374, and others). Also, silk scaffolds have been widely used for meniscus tissue engineering, repair, adhesion, and regeneration by Kaplan Lab and others (Biomaterials, 2011, 32, 2, 639, Theranostics, 2020, 10, 5090, Acta Biomaterialia, 2012, 8, 289, etc). Similarly, methacrylated polymers (such as hyaluronic acid, and gelatin) have also been used as adhesive patch for meniscus repair (Materials and Design, 2023, 229, 111915, International Journal of biological macromolecules, 2018, 110, 479-487 etc). While there is some novelty in the composition of materials and their applications, this novelty is not enough for a publication in a high-impact and prestigious journal like Nature Communication. In my opinion, the article is well-suited for Advanced Healthcare Material and similar journals. Because of the above reasons and the lack of substantial novelty required for the top-tier journal, I would like to reject the paper for publication in Nature Communications.

Other points to further improve the manuscript

1) The authors should provide a reference for the alkylation reaction between 4-bromomethyl phenylboronic acid and 1-vinyl imidazole (lines 111-112)

- 2) It would be great to label the individual proton NMR peaks of SF and SFMA with the corresponding hydrogens (Figure S1A)
- 3) Beta-sheet quantification: Figure 1H and S3A are the same figures and repetitive. The authors should delete the figure in the main figure
- 4) For figure 1i, the authors have demonstrated that phenyl boronic acid is attached to the silk chain, however, did not mention which amino acid of silk interacts with them.
- 5) The authors have demonstrated SEM images of different S-PIL compositions and showed their microporous structures in the hydrogel adhesives. Since pore structure and diameter are important for different biomedical applications, it would be great to know the pore size of each material composition. Also, as secondary structure and networking increase, it is expected to have smaller pore size.
- 6) For all the figures showing statistics, methods for analysis of variance must be mentioned in the figure caption.
- 7) A paragraph on statistical methods used must be added in the materials and method section in the supporting information
- 8) The authors have used IL1 and IL2 as other two ionic liquids as controls to compare their adhesive performance in comparison to PIL. The full chemical name of the ionic liquid should be added and the possible reason of their low adhesion in comparison to PIL must be discussed (less interaction with SFMA and so on)
- 9) Figure 2F, G, I- Scale bar needed.
- 10) The biocompatibility of different material systems (S-Gel and S-PIL) should be performed for at least 1 week (or 2 weeks if possible). The authors have shown only for 5 days. Further, the scale bar for Figures 3c and d are missing.
- 11) For the in vivo compatibility test, the scale bar of the hydrogel is missing (Figure S10). Also, in comparison with the S-Gel, S-PIL10 appears to have more degradation (Figure S10C). What could be the possible reason?
- 12) Reference needed to statement in lines 277-278.
- 13) In lines 280-282, the authors mentioned inefficiency in current treatment (meniscectomy and others). The authors must discuss different material systems used to treat, their limitations, and how the current design is better for meniscus repair. What are the advantages of silk-based material systems over others (Hyaluronic acid, gelatin, and others)?

Dear editors and reviewers,

Thank you very much for your insightful and positive assessment of our manuscript entitled "**(Silk Fibroin Hydrogel Adhesive Enables Sealed-tight Reconstruction of Meniscus Tears)**" (NCOMMS-23-47640A). Based on these valuable comments and suggestions, we have diligently addressed each one and made careful modifications to the original manuscript. All changes made to the manuscript are in red color. We hope our revisions address the reviewers' questions and suggestions and that the revised manuscript meets the standards for publication in Nature Communications.

Thank you very much for your consideration.

Sincerely,

Hongwei Ouyang, M.D., Ph.D.
Zhejiang University, Hangzhou 310058, China

Point-by-point responses to comments from the reviewers:

Reviewer #1

The study presented in this paper is interesting, well carried out and well presented. The conclusions are based on evidences presented herein. This work is without any doubt worth for publication in some journal; the originality is limited to the use of silk fibroin, modified with a well-known procedure (vinyl imidazole) and the tissue engineering is based on well-known growth factor tgf-beta.

Response: We would like to express our great thanks for the reviewer's comprehensive assessment, which is interesting and well-presented. We appreciate the opportunity to re-clarify the significance and innovation of this work. The most significant innovation of this work lay in the revolutionary strategy of meniscus tear therapy and we verified that the concept of meniscus adhesive was feasible.

These two common meniscus tear therapies, like partial or total meniscectomy and defect closure with sutures, have non-negligible drawbacks that limit their success rate of application, and the former would reduce the force

area to bring more load on the cartilage while the latter fails to repair the meniscus in the white-white zone from the point of regeneration. Moreover, the reported meniscal scaffolds like the Menaflex Collagen Meniscus Implant and Actifit scaffolds, were used to substitute the torn meniscus and showed unsatisfactory outcomes, especially without observing their long-term effectiveness. Thus, tissue adhesives arise as a highly practicable method in clinic to repair meniscus tears, as the strategy could not only save the meniscus to preserve the integrity of the native meniscus tissue, but also ensure tight adhesion of the torn meniscus tissue until complete healing. In 1995, cyanoacrylate (the main component of 502 glue) based adhesive was first used for the repair of meniscal tears. Though in vitro experiments showed good performance, this glue cannot be used for meniscal repair in vivo due to its cytotoxicity and severe inflammatory reactions. Then, fibrin, a natural polymer material, was used in the repair of meniscal tears. However, fibrin had weaker mechanical properties and lower adhesive capacity, resulting in poorer effect in repair of meniscal tears. Furthermore, block copolymer bioadhesives based on trimethyl vinyl (TMC) were also used to repair meniscal tears. This hydrogel adhesive had stronger strength than fibrin glue, and the elastic modulus also reached the level of natural meniscus, but biocompatibility still needs to be solved. Actually, these reported adhesives were used to adhere meniscus tears in vitro, and few cases using hydrogel adhesives for the successful repair of meniscus tears in vivo without the assistance of sutures have been reported due to the repair difficulties and the demand for the mechanics.

Hence, we overcame these difficulties about the limited strategy of meniscus tears and showed the first meniscus adhesive used in vivo meniscus tear repair without sutures. From the perspective of materials science, methacrylation of silk has been reported in some journals (Nature Communications, 2018, 9, 1620, Nature Protocol, 2021, 16, 5484, and others), but we have made improvements and developed a functional hydrogel adhesive S-PIL10 composed of glycidyl methacrylate-modified silk fibroin

(SFMA) and original synthesized phenylboronic acid-ionic liquid (PIL). The molecule vinyl imidazole is part of the synthesized PIL, and the structure of tailor-made PIL was equipped with three features: 1) the vinyl group in the imidazole cation acting as a unit to copolymerize with SFMA; 2) the structure of the imidazole salt generating hydrogen bonding and promoting the formation of β -sheet structures for silk fibroin; 3) the phenylboronic acid groups reacting with hydroxy groups in the chain of silk, mainly from tyrosine and serine, to form dynamic boronic ester bonds in the network. Due to these interactions in the hydrogel network, this tissue adhesive had superior wet adhesion properties (shear adhesive strength: 182.12 kPa), swelling resistance (SR: 1.06), and anti-fatigue capabilities (1.96 MPa) (**Fig. 2**), which were comparable to adhesive materials reported in top journals (Science Translational Medicine 2022, 14, eabh2857; Nature Materials 2021, 20, 229-236; Nature 2019, 575, 169-174).

Fig. 2 Adhesion performance and mechanical properties of S-PIL series. (A) Schematic diagrams of lap shear testing on the meniscus slice (B) Load-displacement curves and lap shear strength of S-Gel with the different concentrations of PIL bonding with meniscus slice. (C) Swelling conditions and (D) swelling ratio of S-Gel and S-PIL10 in PBS buffer. (E) Ashby plot of S-PIL10 compared with reported adhesives. (F) Macrograph and stress-strain curves of S-PIL10 with cyclic compressive loading-unloading testing for 1000 cycles. (G) Twisting and water-flushing after meniscus adhesion by S-PIL10. (H) Load-displacement curves of S-PIL gel bonding meniscus tear and SEM image of the interface between S-PIL10 and meniscus. (I) Retention condition of S-PIL bonding meniscus tear after 60 days. Data have been presented as mean \pm SD, n = 3. Statistically significant differences, as analyzed using ANOVA followed by Tukey's multiple comparison test, have been indicated as *p < 0.05, **p < 0.01, ***p < 0.001, and ****p < 0.0001.

Besides, the use of transforming growth factor-beta 1 (TGF- β 1) was based on our growth factor screening from a lot of candidates. In the results of the *in vivo* performance of meniscus tears in this work, the addition of TGF- β 1 can promote the regeneration of meniscus tears in the white-white zone (**Fig. 3**), which is crucial for meniscus repair. Therefore, we have selected the most suitable factor to enhance the therapeutic effect in application of meniscus adhesive. The meniscus adhesive improved inflammatory microenvironment and upregulated meniscus-related gene expressions through the elimination of reactive oxygen species (ROS) and continuous release of TGF- β 1. Combined with its competitive mechanics, the meniscus adhesive presented excellent therapeutic performance *in vivo*. These exciting results motivate us to write the research paper as a Communication. We firmly believe that our work has the potential to offer significant contributions to this field, and meniscus adhesives could become a clinical method for *in-situ* meniscus tears repair with the development of more and more meniscus adhesives in the future.

Fig. 3 Biocompatibility and ex vivo therapeutics of meniscus adhesives. (A) Schematic illustration of co-culture with hydrogels and live/dead cell assays. (B) Cell viability of rabbit meniscus cells co-cultured with hydrogels within 7 days. (C) Cell Counting Kit 8 (CCK-8) assays of rabbit meniscus cells co-cultured with hydrogels within 7 days. (D) Alcian blue (AB) staining and immunofluorescence staining of meniscus cells. (E) Meniscus-related gene (Sox9, Col2a1 and ACAN) expression in the S-Gel and S-PIL10 at 2 weeks. (F) TGF-β1 release-time curves of S-Gel and S-PIL10. (G) Inflammatory

condition after meniscus tears and DDPH scavenging and PTIO scavenging of S-Gel and S-PIL10. Scale bars are 200 μm . Data have been presented as mean \pm SD, n = 3. Statistically significant differences, as analyzed using ANOVA followed by Tukey's multiple comparison test, have been indicated as *p < 0.05, **p < 0.01, ***p < 0.001, and ****p < 0.0001.

Here, we would like to re-emphasize the highlights of this work, and the three major findings are listed below:

1. The most significant innovation of this work lay in the revolutionary strategy of meniscus tear therapy and we verified that the concept of meniscus adhesive was feasible.

2. The hydrogel adhesive S-PIL10 was composed of glycidyl methacrylate-modified silk fibroin (SFMA) and custom-designed phenylboronic acid-ionic liquid (PIL) had superior wet adhesion, swelling resistance, and anti-fatigue capabilities, and could be comparable to these top journals (Science Translational Medicine 14 (2022) eabh2857; Nature Materials 20 (2021) 229-236; Nature 575 (2019) 169-174). Moreover, S-PIL10 was designed based on silk fibroin (natural biomacromolecule) and had good biocompatibility in favor of promoting tissue regeneration.

3. The meniscus adhesive improved inflammatory microenvironment and upregulated meniscus-related gene expressions through the elimination of reactive oxygen species (ROS) and continuous release of TGF- β 1. Combined with its competitive mechanics, the meniscus adhesive was first used in vivo for meniscus tear repair and presented excellent therapeutic performance.

To clarify the novelty more clearly, we have re-edited the following text in the Abstract and Discussion as follows:

“Supramolecular interactions of β -sheets and hydrogen bonds richened by unique phenylboronic acid-ionic liquid (PIL) resulted in enhanced wet adhesion, swelling resistance, and anti-fatigue capabilities, compared to neat silk fibroin gel.”

“*In vivo* rabbit models functionally evidenced the seamless and dense reconstruction of torn meniscus, verifying that the concept of meniscus adhesive was feasible and providing a promising revolutionary strategy for preclinical research to repair meniscus tears.”

“The key purpose of this work lay in the revolutionary strategy of meniscus tear therapy, and was to verify whether the concept of meniscus adhesive was feasible and provide a novel clinical method that could repair various types of meniscus tears due to inefficacy in current treatments. For example, partial or total meniscectomy and defect closure with sutures have non-negligible drawbacks that limit their success rate of application, and the former would reduce the force area to bring more load on the cartilage while the latter fails to repair the meniscus in the white-white zone from the point of regeneration ¹.”

“In 1995, the cyanoacrylate adhesive (the main component of 502 glue) was first used for the repair of meniscal tears. Though *in vitro* experiments showed good performance, this glue cannot be used for meniscal repair *in vivo* due to its cytotoxicity and severe inflammatory reactions ⁵³. Then, fibrin, a natural polymer material, was used in the repair of meniscal tears. However, fibrin had weaker mechanical properties and lower adhesive capacity, resulting in poorer repair of meniscal tears ⁵⁴. Furthermore, block copolymer bioadhesives based on trimethyl vinyl (TMC) were also used to repair meniscal tears. This hydrogel adhesive had stronger strength than fibrin glue, and the elastic modulus also reached the level of natural meniscus ⁵⁵, but biocompatibility still needs to be solved. Actually, these reported adhesives were used to adhere meniscus tears *in vitro*, and few cases using hydrogel adhesives for the successful repair of meniscus tears *in vivo* without the assistance of sutures have been reported due to the repair difficulties and the demand for the mechanics ^{7, 56}. In the

current work, we aimed to develop an instant robust hydrogel adhesive based on SFMA, PIL, and TGF- β 1 to enable sealed-tight reconstruction of meniscus tears.”

Reviewer #2

In this manuscript, Pan et al. present a novel hydrogel adhesive utilizing silk fibroin and phenylboronic acid-ionic liquid (PIL). The material presented here is quite intriguing as it combines the components of a radical scavenger, tissue adhesive, and drug delivery carrier, which are very important in meniscal tear repair and regeneration. The authors have presented a detailed characterization of the synthesized material and the biocompatibility data presented here is also appropriate. Overall, the manuscript is well organized with detailed discussion. I do have the following comments and questions which should be addressed to improve the manuscript.

Response: Thank you very much for your great work reviewing our manuscript. Your comments are very helpful for improving our work. The manuscript has been revised carefully according to your comments. We hope that the revised manuscript will meet with your approval.

Major concerns:

1. The tissue adhesive property is essential for the material in this work. The authors have compared the adhesive results with commercial products and lab-made other types of IL. However, why was this PIL chosen to enhance the properties of the adhesive and the adhesion mechanism should be discussed in detail.

Response: We appreciate the insightful comments from the review. PIL was synthesized and could enhance the properties of the hydrogel adhesive due to these multiple interactions. The structure of tailor-made PIL possessed three strengthening mechanisms: 1) the vinyl group in the imidazole cation as a

monomer copolymerized with SFMA; 2) the structure of the imidazole salt generating hydrogen bonding and promoting the formation of β -sheet structures for silk fibroin; 3) The phenylboronic acid groups reacting with hydroxy groups to form dynamic boronic ester bonds in the network. Besides, the adhesion mechanism has been discussed as follows:

“On the one hand, an in-situ photocurable instant S-PIL10 was fabricated, which fulfilled the gaps between the torn meniscus tissue and formed the mechanical interlocking to improve interfacial adhesion ⁵⁷. Actually, noncovalent interactions play an important role in achieving fast interfacial bonding and contributing to adhesion stability ⁵⁸. S-PIL10 could provide noncovalent interactions in tissue adhesion including electrostatic interactions and hydrogen bonds due to its various active functional groups. On the other hand, the hydrogel adhesive itself was strengthened through selecting robust silk fibroin and designing tailor-made ionic liquid ⁵⁹. The unique ionic liquid reacted with SFMA by multiple interactions, including copolymerization with the vinyl group, production of dynamic boronic ester bonds through the phenylboronic acid groups reacting with hydroxy groups, and the generation of hydrogen bonding and formation of many β -sheet structures through the Hoffmeister effect of PIL on silk fibroin ⁶⁰.”

2. The meniscus adhesive solidified within 5s and the time is short. Does the addition of PIL affect the gelation time of SFMA hydrogel?

Response: Thank you for raising these questions. According to the rheological data (Fig. 1D), we compared the gel time of different hydrogels (**Fig. S2**). After statistical analysis, it was found that PIL did not significantly affect the gel time of the different hydrogels.

“Moreover, the addition of PIL did not significantly affect the gel time of the different hydrogel adhesives (**Fig. S2**).”

Fig. S2 The gelatin time of S-Gel and S-PIL gels. Data have been presented as mean \pm SD, $n = 3$. Statistically significant differences, as analyzed using ANOVA followed by Tukey's multiple comparison test, have been indicated as ns: no significance.

3. It is interesting the hydrogel has anti-swelling properties when using PIL in this work. Can the authors comment on the mechanism of this phenomenon? Can this PIL be used for other materials such as GelMA or PEGDA and lead to the same anti-swelling property?

Response: This is indeed a topic of central importance, and we thank the reviewer for raising the point. In the revised manuscript, we explained the anti-swelling behaviors of different hydrogels. Also, we performed some anti-swelling experiments about GelMA, PEGDA, and HAMA hydrogel with PIL and discussed these results in the manuscript.

“These excellent properties can be attributed to the multiple interactions in the polymeric network, which enhance the mechanical properties to resist swelling. Additionally, the β -sheet structures and PIL in the network are hydrophobic, which contributes to the resistance against water absorption. Besides, studies have shown that the denser and smaller the pores of the polymeric network, the higher the anti-swelling properties of the hydrogel (**Fig. S6**)³⁷. Generally, other materials, such as GelMA and PEGDA, also exhibit similar anti-swelling performance due to the incorporation of PIL (**Fig. S9&S10**). Notably, the anti-

swelling properties of HAMA hydrogel were significantly improved, further verifying the superiority of tailor-made PIL.”

Fig. S6 Pore size of S-Gel and S-PIL gels in SEM images. Data have been presented as mean \pm SD, n = 6.

Fig. S9 Swelling conditions of GeIMA, PEGDA and HAMA with PIL in PBS buffer. Scale bar: 10 mm.

Fig. S10 Swelling ratio of GeIMA, PEGDA, and HAMA with PIL in PBS buffer. Data have been presented as mean \pm SD, n = 3.

4. S-PIL10 has good compression performance. However, the energy dissipation is also important for the meniscus. The energy dissipation of S-PIL10 needs to be further demonstrated in comparison with S-Gel during loading-unloading tests for 1000 cycles.

Response: Many thanks for the excellent suggestion. We added **Fig. S12** to illustrate the energy dissipation of S-PIL10 during the loading-unloading compression test. S-PIL10, like most meniscus hydrogels, exhibits a decline in energy dissipation with increasing compression cycles.

“And with the increasing compression cycles, S-PIL10 exhibits a decline in energy dissipation (**Fig. S12**), which was similar to meniscus and its derived materials.”

Fig. S12 The energy dissipation during loading-unloading tests for 1000 cycles.

5. The authors demonstrated the antioxidant activity of the material using DPPH and PITO assay. How about the antioxidant activity in vitro using cells?

Response: Thank you for your valuable feedback on our manuscript. The antioxidant activity of hydrogel at the cellular level needs to be further confirmed. The experiments have been performed and the results have been discussed in the revised manuscript.

“Intracellular antioxidant ability of hydrogels

To investigate the intracellular oxidative stress protection of the hydrogels, we used a ROS probe DCFH-DA to assess intracellular ROS levels. Typically, L929 fibroblasts were seeded at a density of 1.5×10^4 cells per well in confocal dishes and incubated for 12 hours. The plates were then reinjected with a fresh medium containing the same volume of either H_2O_2 or H_2O_2 with hydrogel treatment. Next, cells were stained with DCFH-DA for 30 minutes according to the manufacturer's instructions. Finally, a confocal laser scanning microscope (A1, Nikon, Japan) was used to observe DCFH-DA fluorescence in the cells.”

“To further confirm the antioxidant activity of hydrogel at the cellular level, the cell protection efficiency of prepared hydrogels by clearing ROS was evaluated through the DCFH-DA kit. As shown in **Fig. S17&S18**, the S-PIL10 group exhibited the weakest green fluorescence and their lowest intracellular ROS levels, suggesting that S-PIL10 had prominent ROS scavenging activity and could protect the cells from being oxidatively damaged *in vitro*.”

Fig. S17 Intercellular ROS scavenging capability of hydrogels evaluated by DCFH-DA (green fluorescence). (Scale bar: 50 μ m)

Fig. S18 Relative DCFH-DA fluorescence ratio of L929 cells treated with hydrogels. Data are presented as means \pm standard deviation (n=3).

6. In histology results, the selection of cartilage positions in the tibia seems inconsistent, and the same location should be chosen for comparison.

Response: Thank you for your suggestions. We have performed experiments and chosen the same tibial position to compare the degree of wear of the tibial cartilage. These changes have been presented in **Fig. 4D** and **Fig. S19B**.

Fig. 4 Repair effect of meniscus tears and evaluation of *in vivo* articular cartilage wear after 8 weeks *in vivo*. (A) Process of meniscus tears modeling and adhesives repair: step 1: transection of the medial collateral ligament; step 2: exposure of meniscus toward the femur; step 3: incision of the meniscus through the full thickness; step 4. In situ sealing by S-PIL adhesives; step 5: solidification by UV and wound closure. (B) Macrograph of meniscus after meniscus radial tears of two months (C) Hematoxylin & eosin (H&E) staining and immunofluorescence on collagen I and collagen II of repaired meniscus with radial tears. (D) Safranin O/fast green staining (SO) of femoral condyles (FCs) and the tibial plateaus (TPs) after meniscus radial tears. Scale bars in (B) are 5 mm, in the first row of (C) are 1mm, and others are 200 μ m.

Fig. S19 (A) Normalized tear area after meniscus tear repair for two months. (B) H&E staining of femoral condyles (FCs) and the tibial plateaus (TPs) after meniscus radial tears. Scale bars are 200 μ m. (C) Osteoarthritis Research Society International (OARSI) scores of femoral condyles (FCs) and the tibial plateaus (TPs) after meniscus tear of two months. Data are presented as means \pm standard deviation (n=3).

Minor comments:

1. In section Adhesion performance, the authors stated they used a standard test method. Strictly, modified methods were used and should be revised because there are differences between the authors' methods and standard methods.

Response: We thank the reviewer for raising this point. Actually, the standard methods were modified to meet our experimental requirements based on our specific application, and relative contents were added in the revised manuscript. "Hence, we moved to the experimental evaluation of the adhesion performance according to the modified lap shear strength test (ASTM F2255) presented in **Fig. 2A.**"

2. “As shown in Fig. 2C, S-PIL10 retained original shape and volume after 14 days, while S-Gel presented significant swelling after only 4 hours, and the swelling ratio decreased from 2.911 to 1.02.”. This statement is confusing and should be clearer.

Response: Thank you for your careful work. We have revised the confusing statement in the new manuscript.

“As shown in Fig. 2C, S-PIL10 retained its original shape and volume after 14 days, while S-Gel presented significant swelling after only 4 hours. Correspondingly, the swelling ratio decreased from 2.911 (S-Gel) to 1.02 (S-PIL10).”

3. Please rewrite “Differing from soft tissue in skin, lung, gastrointestinal tract or cardiovascular tissues, the design of adhesives for meniscus-like hard tissues necessarily integrated satisfied adhesive strength, robust mechanical properties as well as long-term regeneration function” in the introduction section, it is confusing.

Response: Thanks for pointing it out. We agree with your comments and the adhesives designed for soft tissue and hard tissues have some differences and should be considered throughout, and then we rewrite this content in the revised manuscript.

“Differing from the tissue adhesives used for soft tissue in skin, lung, gastrointestinal tract or cardiovascular tissues, the design of adhesives for meniscus-like hard tissues necessarily integrated satisfied adhesive strength, robust mechanical properties as well as long-term regeneration function.”

4. In the section ‘Biocompatibility and ex vivo therapeutics of meniscus adhesives’, Did the authors perform studies directly seeded cells on the surface of the adhesives or within the adhesives? Why did S-PIL15 have a slight inhibition on the proliferation of rabbit meniscus cells? Does it concentration dependent?

Response: Thanks for the query. Considering the actual in vivo condition, cells were cultured on the surface of the hydrogel adhesives to evaluate their biocompatibility. S-PIL15 has a light inhibition on the proliferation of rabbit meniscus cells, as PIL itself may be not very friendly to cells, and with the increase of the concentration, S-PIL series would have negative effect on the cells. PIL was added into the hydrogel adhesive and had multiple interactions in the polymeric network to prevent the leakage of PIL. Therefore, within certain concentrations, S-PIL10 was considered as the balanced formulation with superior mechanics and cytocompatibility, which was optimally selected in the following meniscus adhesive study. These relevant contents have been added in the revised manuscript.

“However, S-PIL15 had a slight inhibition on the proliferation of rabbit meniscus cells, as PIL itself may be not very friendly to cells, and with the increase of the concentration, S-PIL series would have negative effect on the cells. PIL was added into the hydrogel adhesive and had multiple interactions in the polymeric network to prevent the leakage of PIL. Integrating the above results, S-PIL10 was considered as the balanced formulation with superior mechanics and cytocompatibility, which was optimally selected in the following meniscus adhesive study.”

5. In the section ‘Biocompatibility and ex vivo therapeutics of meniscus adhesives’, Is there any difference in inflammatory response among groups? I cannot see from the low mag image. How about the chronic inflammation?

Response: Thanks for the valuable questions. According to the reviewer’s suggestion, we changed a higher mag image (**Fig. S15**) to observe the inflammation. Silk fibroin is a well-known biomaterial with low immunogenicity and there is almost no significant difference in inflammatory response among S-Gel and S-PIL10 groups. With the passage of time, the acute inflammation disappeared and the chronic inflammation was almost invisible, suggesting that the hydrogel adhesive and its degradation product have good biocompatibility.

The acute and chronic inflammation have been discussed specifically in the revised manuscript.

“Silk fibroin is a well-known biomaterial with low immunogenicity and there was almost no significant difference in inflammatory response among S-Gel and S-PIL10 groups. Inflammatory cells were observed around the implants in the first week and gradually disappeared over time (**Fig. S15B**), the acute inflammation disappeared and the chronic inflammation was almost invisible, suggesting that the hydrogel adhesive and its degradation product have good biocompatibility.”

Fig. S15 (A) Schematic illustration of Subcutaneous *in vivo* degradation. (B) Macrograph and Hematoxylin & eosin (H&E) staining images of implanted S-Gel and S-PIL10 with the surrounding skin at different time points. (C) The mass remained of S-Gel and S-PIL10.

Data are presented as means \pm standard deviation (n=3). (Scale bars at 1 and 4 rows are 5 mm, Scale bars at 2 and 5 rows are 1 mm, Scale bars at 3 and 6 rows are 50 μ m)

6. In the section 'Biocompatibility and ex vivo therapeutics of meniscus adhesives', "... , probably due to a slight degradation of the hydrogel adhesives. Notably, the structure of S-PIL10 at week 2 was cracked and significant cells proliferated between inter-hydrogel space, indicating the ideal biocompatibility and repair potentials of S-PIL10 (Fig. S10C)...". From the macroscopic images, there is slight shrinkage in adhesive size by 10 weeks, indicating that these adhesives might have relatively slow degradation rates. What is the optimal degradation of adhesive for this application? Did the authors perform any degradation studies?

Response: Thank you for raising these constructive questions. Ideally, the optimal degradation of our hydrogel adhesive in this meniscus should match the rate of meniscus regeneration. However, achieving the perfect rate in reality is challenging, and we are pleased to report that after 8 weeks, the meniscus tears had sealed-tight repair without any adhesive residue after using our instant robust hydrogel adhesive, suggesting that the adhesive and its degradation product didn't inhibit meniscus regeneration and the rate of meniscus regeneration could keep up with the degradation rate of adhesive. As reported, silk fibroin degrades in vitro in response to proteolytic enzymes. Considering that the hydrogel adhesive was used in vivo, the degradation studies were performed subcutaneously, and we found these adhesives degraded slowly after 8 weeks, which was consistent with previously reported studies. We also wanted to perform degradation studies in situ (meniscus tears), but the data were hard to obtain due to the minimal amount of hydrogel adhesive used in the meniscus (<20 μ L). Besides, the meniscus repaired by the adhesive endured compressive stress and friction, which facilitated the degradation of the hydrogel adhesive. Overall, the meniscus adhesive was used in vivo meniscus tears repair with a slow degradation rate and presented

excellent therapeutic performance. These relevant contents have been added in the revised manuscript.

“Further, *in vivo* biocompatibility and degradation test of meniscus adhesives were performed subcutaneously (**Fig. S15A**).”

“And the morphology of S-PIL 10 showed mild degradation, but the weight of S-Gel changed significantly within 8 weeks (**Fig. S15C**). S-Gel first experienced significant swelling with the maximal size achieved at day 3 and then shrank in 8 weeks, and the mass of S-PIL10 *in vivo* increased slightly in the beginning, possibly caused by the rising of water content, consistent with previous swelling results. Subsequently, the mass of S-Gel and S-PIL10 was reduced, probably due to a slight degradation of the hydrogel adhesives. During the degradation process of S-Gel and S-PIL10, the swelling behavior of hydrogels could not be ignored. Compared with S-PIL10, S-Gel has more body fluid in the polymeric network, which contributes to the higher mass remaining of S-Gel, and the mass remaining of S-PIL10 was lighter with less water due to its excellent anti-swelling properties. These factors jointly resulted in the observed differences in the degradation process between the two hydrogels.”

7. In the section ‘Biocompatibility and ex vivo therapeutics of meniscus adhesives’, “Also, TGF- β 1 was released more rapidly in S-Gel compared with the release performance in S-PIL10 due to the addition of PIL”. It is better to state that S-PIL had slow release. Does TGF- β 1 cross-linked with other components during the UV treatment?

Response: Thanks for the careful review and the query. We have taken the reviewer’s suggestions and rephrased certain sections. For the second question, TGF- β 1 is composed of amino acids, and could potentially interact with these hydrogels through noncovalent bonds, thereby limiting their release from the hydrogel adhesives, but these interactions do not change the properties of TGF- β 1. These relative contents have been revised in the manuscript.

“The release curve of TGF- β 1 loaded S-Gel and S-PIL10 demonstrated sustained release in more than 8 weeks (**Fig. 3F**) due to the noncovalent interactions between TGF- β 1 and these polymeric networks. Also, TGF- β 1 was released more slowly in S-PIL10 compared with the release performance in S-Gel due to the addition of PIL, which could be explained by anti-swelling ability of S-PIL10 and the smaller size of pores compared with S-Gel.”

8. In the section ‘The application of S-PIL10 on in vivo multiple types of meniscus tears’, the authors stated that “Besides, S-PIL10+GF group had the smallest area of tear based on statistical results of histological staining (Fig. S12A)”. Given the smallest area of tear, do the authors expect whether it will completely repair the tear in a longer time or not?

Response: We appreciate the reviewer’s guidance on this point. As presented in **Fig. 4B**, macroscopic views of the rabbit meniscus showed no significant tear in the S-PIL10+GF group. While there was a small area of tear visible in H&E staining and immunofluorescence staining of the meniscus, it may have no effect on the free movement of rabbits. We also expected it could completely repair the tear in a longer time, but we didn’t find the residual hydrogel adhesive after 2 months and the meniscus could not be self-healed in white-white zone. In this manuscript, our goal with these results was to demonstrate that the strong adhesion performance of S-PIL10 can adhere to the meniscus tears tightly and the addition of TGF- β 1 can promote the regeneration of meniscus tears in the white-white zone. As you commented above, to have excellent and perfect therapeutic performance, the meniscus adhesive should be engineered with clinical trials.

9. Some of the methods are not completed and should be added in Supplementary Information.

Response: Thank you for your careful work. We have added the methods and revised in the Supplementary Information.

“The femoral condyles (FCs) and the tibial plateaus (TPs) were blindly and independently evaluated by three researchers according to the Osteoarthritis Research Society International (OARSI) osteoarthritis cartilage histopathology assessment system, in which a higher score suggested worse cartilage condition.”

“Statistical analysis

All experiments were conducted at least three times independently, and the results were expressed as mean±SD. All statistical analysis was performed using origin and graph pad prism software. Statistical analysis was performed using one-way ANOVA followed by Tukey’s test, n=3. Data have been presented as mean ± SD, n = 3. Statistically significant differences, as analyzed using A NOVA followed by Tukey's multiple comparison test, have been indicated as *p < 0.05, **p < 0.01, and ***p < 0.001, ns: no significance. A notation of NS (not significant) was used when P values were greater than 0.05.”

Reviewer #3

In this manuscript by Pan et al., the authors have designed a material system based on silk fibroin that is composed of methacrylated silk crosslinked with phenylboronic acid-ionic liquid loading with growth factor TGF-β1, which exhibits instant gelation, demonstrates strong adhesion, swelling resistance, and anti-fatigue capabilities, compared to neat silk fibroin gel. Further, the authors used them to repair the meniscus as meniscus adhesive which facilitates healing by scavenging reactive oxygen species and upregulation of different genes. The material system is not completely novel as methacrylation of silk is widely reported (Nature Protocol, 2021, 16, 5484, ACS Biomater Sci. Eng., 2019, 5, 12, 6374, and others). Also, silk scaffolds have been widely used for meniscus tissue engineering, repair, adhesion, and regeneration by Kaplan Lab and others (Biomaterials, 2011, 32, 2, 639, Theranostics, 2020, 10, 5090, Acta Biomaterialia, 2012, 8, 289, etc). Similarly, methacrylated polymers (such as hyaluronic acid, and gelatin) have also been used as adhesive patch for

meniscus repair (Materials and Design, 2023, 229, 111915, International Journal of biological macromolecules, 2018, 110, 479-487 etc). While there is some novelty in the composition of materials and their applications, this novelty is not enough for a publication in a high-impact and prestigious journal like Nature Communication. In my opinion, the article is well-suited for Advanced Healthcare Material and similar journals. Because of the above reasons and the lack of substantial novelty required for the top-tier journal, I would like to reject the paper for publication in Nature Communications.

Response: We thank the reviewer for the valuable comments. We understand the concerns of the reviewer and would like to clarify the significance and innovation of this work. The most significant innovation of this work lay in the revolutionary strategy of meniscus tear therapy and we verified that the concept of meniscus adhesive was feasible.

Methacrylation of silk was first reported in Nature Communications, 2018, 9, 1620, and then has been widely used in different research (Nature Protocol, 2021, 16, 5484; Appl. Mater. Today 2021, 23; 101004, ACS Biomater. Sci. Eng., 2019, 5, 12, 6374, and others). Based on the above ideas and perspectives, we have made improvements and developed a hydrogel adhesive S-PIL10 composed of glycidyl methacrylate-modified silk fibroin (SFMA) and original synthetic phenylboronic acid-ionic liquid (PIL). The structure of tailor-made PIL possesses three features: 1) the vinyl group in the imidazole cation as a monomer copolymerized with SFMA; 2) the structure of the imidazole salt generating hydrogen bonding and promoting the formation of β -sheet structures for silk fibroin; 3) The phenylboronic acid groups reacting with hydroxy groups to form dynamic boronic ester bonds in the network. Based on these multiple interactions in the hydrogel network, this tissue adhesive had superior wet adhesion (shear adhesive strength: 182.12 kPa), swelling resistance (SR: 1.06), and anti-fatigue capabilities (1.96 MPa), which were comparable to these top

journals (Science Translational Medicine 2022, 14, eabh2857; Nature Materials 2021, 20, 229-236; Nature 2019, 575, 169-174).

More importantly, the focus of this paper is also to demonstrate the innovative application in the field of meniscus repair. The utilization and modification of current biomaterials for new medical applications offer equivalently significant innovative and clinical translational value. The meniscus adhesive was first used in vivo meniscus tears repair without sutures and presented excellent therapeutic performance. Compared with some reported adhesive patches for meniscus repair (Materials and Design, 2023, 229, 111915, International Journal of biological macromolecules, 2018, 110, 479-487 etc), our meniscus adhesive could repair the meniscus tears in the white-white zone (defect closure with sutures fails to repair the meniscus in this zone from the point of regeneration). Besides, the strategy of silk scaffolds (Biomaterials, 2011, 32, 2, 639, Theranostics, 2020, 10, 5090, Acta Biomaterialia, 2012, 8, 289, etc) was completely different from meniscus adhesives, and the former was used to substitute the torn meniscus while the latter was used to achieve closure of meniscus tears. When the meniscus was severely damaged, the meniscus scaffold would be chosen; but in most milder cases of meniscus tears, the meniscus adhesives could be a better option to replace sutures. We firmly believe that our work has the potential to offer significant contributions to this field, and meniscus adhesives could become a clinical method for in-situ meniscus tears repair with the development of more and more meniscus adhesives in the future.

Here we would like to re-emphasize the highlights of this work, and the three major findings are listed below:

1. The most significant innovation of this work lay in the revolutionary strategy of meniscus tear therapy and we verified that the concept of meniscus adhesive was feasible.

2. The hydrogel adhesive S-PIL10 was composed of glycidyl methacrylate-modified silk fibroin (SFMA) and custom-designed phenylboronic acid-ionic liquid (PIL) had superior wet adhesion, swelling resistance, and anti-fatigue capabilities, and could be comparable to these top journals (Science Translational Medicine 14 (2022) eabh2857; Nature Materials 20 (2021) 229-236; Nature 575 (2019) 169-174). Moreover, S-PIL10 was designed based on silk fibroin (natural biomacromolecule) and had good biocompatibility in favor of promoting tissue regeneration.

3. The meniscus adhesive improved inflammatory microenvironment and upregulated meniscus-related gene expressions through the elimination of reactive oxygen species (ROS) and continuous release of TGF- β 1. Combined with its competitive mechanics, the meniscus adhesive was first used in vivo meniscus tears repair and presented excellent therapeutic performance.

To strengthen the novelty, we have re-edited the following text in the Introduction and Discussion as follows:

“The key purpose of this work lay in the revolutionary strategy of meniscus tear therapy, and was to verify whether the concept of meniscus adhesive was feasible and provide a novel clinical method that could repair various types of meniscus tears due to inefficacy in current treatments. For example, partial or total meniscectomy and defect closure with sutures have non-negligible drawbacks that limit their success rate of application, and the former would reduce the force area to bring more load on the cartilage while the latter fails to repair the meniscus in white-white zone from the point of regeneration ¹.”

“Thus, tissue adhesives indicated a highly clinical therapeutic method to repair meniscus tears, as the strategy saved the meniscus to preserve the integrity of the native meniscus tissue, and ensured tight adhesion of the torn meniscus tissue until complete healing.”

“The unique ionic liquid reacted with SFMA by multiple interactions, including copolymerization with the vinyl group, production of dynamic boronic ester bonds through the phenylboronic acid groups reacting with hydroxy groups, and the generation of hydrogen bonding and formation of many β -sheet structures through the Hoffmeister effect of PIL on silk fibroin ⁶⁰.”

“Equipped with these superior properties, the hydrogel adhesive S-PIL10 presented excellent repairing performance in various types of meniscus tears of New Zealand white rabbits and demonstrated that the strong adhesion properties of S-PIL10 and the function of TGF- β 1 complemented each other in the repair process.”

The manuscript has been revised carefully according to your comments. We hope that the revised manuscript will meet with your approval.

Other points to further improve the manuscript

1) The authors should provide a reference for the alkylation reaction between 4-bromomethyl phenylboronic acid and 1-vinyl imidazole (lines 111-112).

Response: Thanks for your careful work. We have provided a reference for the key alkylation reaction in the revised manuscript.

“PIL was synthesized through an alkylation reaction between 4-(bromomethyl) phenylboronic acid and 1-vinyl imidazole ²² (**Fig. 1A**).”

2) It would be great to label the individual proton NMR peaks of SF and SFMA with the corresponding hydrogens (Figure S1A).

Response: Thank you for your helpful advice. In the revised version, we have labeled the individual proton NMR peaks of SF and SFMA to make the results clearer.

“Their typical peaks were consistent with their chemical structures and functional groups, representing $\delta = 6.0$ ppm and 5.6 ppm in SFMA spectrum for methacrylate vinyl group signals ²⁴.”

Fig. S2 (A) ¹H-NMR spectra of silk fibroin (SF) and methacrylated silk fibroin (SFMA). (B) FTIR spectrum of SF and SFMA. (C) FTIR spectrum of PIL. (D) FTIR spectrum of S-Gel and S-PIL gels.

3) Beta-sheet quantification: Figure 1H and S3A are the same figures and repetitive. The authors should delete the figure in the main figure.

Response: Thanks for your constructive advice. We are sorry to use the same color to cause the misunderstanding. **Figure 1H** is the β -sheet quantification of S-PIL10, and **Figure S4A** is the β -sheet quantification of S-Gel, but they are in the same yellow color. We have corrected this in the revised manuscript (the color of Figure S4A was changed).

Fig. S4 Quantitative analysis of secondary structures of (A) S-Gel, (B) S-PIL5 and (C) S-PIL15.

4) For figure 1i, the authors have demonstrated that phenyl boronic acid is attached to the silk chain, however, did not mention which amino acid of silk interacts with them.

Response: Thank you for your valuable suggestions. There are abundant hydroxy groups in the chain of silk, mainly from tyrosine and serine, and the phenylboronic acid groups react with hydroxy groups to form dynamic boronic ester bonds in the polymeric network. We have clarified this in the revised manuscript.

“The phenylboronic acid groups reacting with hydroxy groups in the chain of silk, mainly from tyrosine and serine, to form dynamic boronic ester bonds in the network.”²³

5) The authors have demonstrated SEM images of different S-PIL compositions and showed their microporous structures in the hydrogel adhesives. Since pore structure and diameter are important for different biomedical applications, it would be great to know the pore size of each material composition. Also, as secondary structure and networking increase, it is expected to have smaller pore size.

Response: Thank you for raising the instructive questions. The pore structure and size are significant for hydrogels because they could provide a favorable space for cell growth and exchange of nutrition. Thus, the average diameter of pores in the SEM images was calculated using ImageJ software and these results were presented in Fig. S5. As expected, with the increase of PIL content, the hydrogels exhibited smaller pore size due to the formation of secondary structure and networks.

“Moreover, SEM images (**Fig. 1J** and **S5**) visualized the microporous structures in these hydrogel adhesives, which could provide a favorable space for cell

growth and exchange of nutrition²⁹. With the increase of PIL amount, the pore size becomes smaller (from 52.53 to 32.59 μm) in **Fig. S6**, which could be explained that more PIL would have more interactions with SFMA and form more secondary structures and networks.”

Fig. S6 Pore size of S-Gel and S-PIL gels in SEM images. Data have been presented as mean \pm SD, n = 6.

6) For all the figures showing statistics, methods for analysis of variance must be mentioned in the figure caption.

Response: Thank you very much for your careful inspection. We are sorry for the omission of the methods for analysis of variance. We have added these methods in the figure caption of the manuscript.

“Data are presented as mean \pm SD, n = 3. Statistically significant differences, as analyzed using ANOVA followed by Tukey’s multiple comparison test, have been indicated as *p < 0.05, **p < 0.01, ***p < 0.001, and ****p < 0.0001, ns: no significance.”

7) A paragraph on statistical methods used must be added in the materials and method section in the supporting information.

Response: Thank you for your good advice. In the revised supporting information, we have added the statistical methods.

“Statistical analysis

All experiments were conducted at least three times independently, and the results were expressed as mean \pm SD. All statistical analysis was performed using origin and graph pad prism software. Statistical analysis was performed using one-way ANOVA followed by Tukey's test, $n = 3$. Data have been presented as mean \pm SD, $n = 3$. Statistically significant differences, as analyzed using ANOVA followed by Tukey's multiple comparison test, have been indicated as * $p < 0.05$, ** $p < 0.01$, *** $p < 0.001$, and **** $p < 0.0001$, ns: no significance. A notation of NS (not significant) was used when P values were greater than 0.05."

8) The authors have used IL1 and IL2 as other two ionic liquids as controls to compare their adhesive performance in comparison to PIL. The full chemical name of the ionic liquid should be added and the possible reason of their low adhesion in comparison to PIL must be discussed (less interaction with SFMA and so on).

Response: Thank you for raising this valuable advice. The full chemical names of the ionic liquids are 1-ethyl-3-methylimidazolium bromide (IL1) and 1-vinyl-3-ethylimidazolium bromide (IL2), and the chemical structures are shown in **Fig. S6B**. Compare with our tailor-made PIL, IL1 didn't possess the vinyl group and IL2 didn't have the phenylboronic acid groups. Thus, the former couldn't act as a monomer copolymerized with SFMA and the latter couldn't react with the hydroxy groups in the hydrogel network. Overall, fewer interactions with SFMA caused their lower adhesion in comparison to PIL, and relative contents were added in the revised manuscript.

"Two other ionic liquids (1-ethyl-3-methylimidazolium bromide and 1-vinyl-3-ethylimidazolium bromide named IL1 and IL2) similarly enhanced the adhesive shear strength when added to the SFMA hydrogel (**Fig. S8B, C, and D**), however, the final adhesive ability was still lower than that of S-PIL10, highlighting the superiority of tailor-made PIL, which could be attributed to their

fewer interactions with SFMA, because IL1 didn't possess the vinyl group and IL2 didn't have the phenylboronic acid groups in comparison with PIL."

9) Figure 2F, G, I- Scale bar needed.

Response: Thanks for your careful work. We are sorry for the missing Scale bar and have corrected it in the Figure 2 of the revised manuscript.

Fig. 2 Adhesion performance and mechanical properties of S-PIL series. (A) Schematic diagrams of lap shear testing on the meniscus slice (B) Load-displacement curves and lap shear strength of S-Gel with the different concentrations of PIL bonding with meniscus slice. (C) Swelling conditions and (D) swelling ratio of S-Gel and S-PIL10 in PBS buffer. (E) Ashby plot of S-PIL10 compared with reported adhesives. (F) Macrograph and stress-strain curves of S-PIL10 with cyclic compressive loading-unloading testing for

1000 cycles. (G) Twisting and water-flushing after meniscus adhesion by S-PIL10. (H) Load-displacement curves of S-PIL gel bonding meniscus tear and SEM image of the interface between S-PIL10 and meniscus. (I) Retention condition of S-PIL bonding meniscus tear after 60 days. Data have been presented as mean \pm SD, n = 3. Statistically significant differences, as analyzed using ANOVA followed by Tukey's multiple comparison test, have been indicated as *p < 0.05, **p < 0.01, ***p < 0.001, and ****p < 0.0001.

10) The biocompatibility of different material systems (S-Gel and S-PIL) should be performed for at least 1 week (or 2 weeks if possible). The authors have shown only for 5 days. Further, the scale bar for Figures 3c and d are missing.

Response: Thanks for your good comments. To make the data more solid, the in vitro cell experiments about S-Gel and S-PIL have been performed for 1 week, and rabbit meniscus cells and L929 fibroblasts both exhibited high cell viability. Besides, the scale bars have been added and all scale bars are 200 μ m in the figure caption. The relative contents were added in the revised manuscript.

“As presented in **Fig.3 A-C**, rabbit meniscus cells maintained good cell viability and apparent proliferation co-cultured with S-PIL series. On Day 7, each well in the 24-well plate was overspread with rabbit meniscus cells. Similarly, L929 fibroblasts also exhibited high cell viability (>95%) on the surface of S-PIL series **(Fig. S14).**”

“The cells were observed through the live/dead staining on Day 1, Day 3 and Day 7.” and “The CCK-8 assay was performed to determine the cell viability at Day 1, Day 3, and Day 7.”

Fig. 3 Biocompatibility and ex vivo therapeutics of meniscus adhesives. (A) Schematic illustration of co-culture with hydrogels and live/dead cell assays. (B) Cell viability of rabbit meniscus cells co-cultured with hydrogels within 7 days. (C) Cell Counting Kit 8 (CCK-8) assays of rabbit meniscus cells co-cultured with hydrogels within 7 days. (D) Alcian blue (AB) staining and immunofluorescence staining of meniscus cells. (E) Meniscus-related gene (Sox9, Col2a1 and ACAN) expression in the S-Gel and S-PIL10 at 2 weeks. (F) TGF-β1 release-time curves of S-Gel and S-PIL10. (G) Inflammatory condition

after meniscus tears and DDPH scavenging and PTIO scavenging of S-Gel and S-PIL10. Scale bars are 200 μm . Data have been presented as mean \pm SD, $n = 3$. Statistically significant differences, as analyzed using ANOVA followed by Tukey's multiple comparison test, have been indicated as * $p < 0.05$, ** $p < 0.01$, *** $p < 0.001$, and **** $p < 0.0001$.

Fig. S14 (A) Live/dead cell assays of L929 fibroblasts on the surface of hydrogels (live cells in green and dead cells in red). (B) Cell viability of L929 fibroblasts on the hydrogels within 7 days. Data are presented as means \pm standard deviation ($n=3$). Scale bar: 200 μm .

11) For the in vivo compatibility test, the scale bar of the hydrogel is missing (Figure S10). Also, in comparison with the S-Gel, S-PIL10 appears to have more degradation (Figure S10C). What could be the possible reason?

Response: Thank you very much for the constructive suggestion. We have added the scale bar in Figure S. Besides, S-PIL10 appears to have more degradation compared with S-Gel from the terms of mass remaining (Figure S10C). To explain the phenomenon, we'd like to first clarify the degradation process. The morphology of S-PIL10 showed mild degradation, but the weight of S-Gel changed significantly within 8 weeks. S-Gel first experienced significant swelling with the maximal size achieved at day 3 and then shrank in 8 weeks, and the mass of S-PIL10 in vivo increased slightly in the beginning, possibly caused by the rising of water content, consistent with previous swelling results. Subsequently, the mass of S-Gel and S-PIL10 was reduced, probably due to a slight degradation of the hydrogel adhesives. "During the degradation process of S-Gel and S-PIL10, the swelling behavior of hydrogels could not be

ignored. Compared with S-PIL10, S-Gel has more body fluid in the polymeric network, which contributes to the higher mass remaining of S-Gel, and the mass remaining of S-PIL10 was lighter with less water due to its excellent anti-swelling properties. These factors jointly resulted in the observed differences in the degradation process between the two hydrogels.” These relevant contents have been added to the revised manuscript.

Fig. S15 (A) Schematic illustration of Subcutaneous *in vivo* degradation. (B) Macrograph and Hematoxylin & eosin (H&E) staining images of implanted S-Gel and S-PIL10 with the surrounding skin at different time points. (C) The mass remained of S-Gel and S-PIL10. Data are presented as means \pm standard deviation (n=3). (Scale bars at 1 and 4 rows are 5 mm, Scale bars at 2 and 5 rows are 1 mm, Scale bars at 3 and 6 rows are 50 μ m).

12) Reference needed to statement in lines 277-278.

Response: Thanks for your careful work. The reference is indeed important and has been added in the revised manuscript.

“Meniscus plays unique roles in the knee joint enabling load transmission, stability and lubrication ⁴⁹.”

13) In lines 280-282, the authors mentioned inefficiency in current treatment (meniscectomy and others). The authors must discuss different material systems used to treat, their limitations, and how the current design is better for meniscus repair. What are the advantages of silk-based material systems over others (Hyaluronic acid, gelatin, and others)?

Response: Thank you so much for your great efforts in reviewing our manuscript. Following your suggestions, different methods and systems used to treat meniscus tears and their limitations have been added in the revised manuscript. Compared with these limited methods, tissue adhesives indicated a highly clinical therapeutic method to repair meniscus tears, as the strategy saved the meniscus to preserve the integrity of the native meniscus tissue, and ensured tight adhesion of the torn meniscus tissue until complete healing. Thus, the adhesive properties are the most important in the sealed-tight reconstruction of meniscus tears and the mechanical properties of hydrogel adhesive itself should be superior. Silk fibroin is a typical natural biomacromolecule with favorable characteristics for the formation of β -sheets and excellent mechanical properties in comparison with hyaluronic acid, gelatin, and others. Hence, we chose silk-based material systems, and the designed hydrogel adhesive S-PIL10 composed of glycidyl methacrylate-modified silk fibroin (SFMA) and own synthetic phenylboronic acid-ionic liquid (PIL) had superior wet adhesion, swelling resistance, and anti-fatigue capabilities, which were comparable to these top journals (Science Translational Medicine 14 (2022) eabh2857; Nature Materials 20 (2021) 229-236; Nature 575 (2019) 169-174), which could be used in vivo meniscus tears repair and presented

excellent therapeutic performance. These relative contents have been added in the revised manuscript.

“In 1995, the cyanoacrylate adhesive (the main component of 502 glue) was first used for the repair of meniscal tears. Though *in vitro* experiments showed good performance, this glue cannot be used for meniscal repair *in vivo* due to its cytotoxicity and severe inflammatory reactions⁵³. Then, fibrin, a natural polymer material, was used in the repair of meniscal tears. However, fibrin had weaker mechanical properties, and lower adhesive capacity, resulting in poorer repair of meniscal tears⁵⁴. Furthermore, block copolymer bioadhesives based on trimethyl vinyl (TMC) were also used to repair meniscal tears. This hydrogel adhesive had stronger strength than fibrin glue, and the elastic modulus also reached the level of natural meniscus⁵⁵. However, biocompatibility still needs to be solved. Actually, these reported adhesives were used to adhere meniscus tears *in vitro*, and few cases using hydrogel adhesives for the successful repair of meniscus tears *in vivo* without the assistance of sutures have been reported due to the repair difficulties and the demand for the mechanics^{7, 56}. In the current work, we aimed to develop an instant robust hydrogel adhesive based on SFMA, PIL and TGF- β 1 to enable sealed-tight reconstruction of meniscus tears.”

Reviewers' Comments:

Reviewer #1:

Remarks to the Author:

The authors have thoroughly and adequately answered to each question. The paper could thus be published as it is.

Reviewer #2:

Remarks to the Author:

Thanks to the authors for addressing the concerns, please carefully check out the manuscript for any grammar errors.

Reviewer #3:

Remarks to the Author:

The authors have comprehensively revised the manuscript with special focus and adding sections on the specific novelties of the current manuscript. In addition, the authors have responded fully to all my comments and other reviewers comments and questions and added them to the manuscript.

I believe the manuscript is significantly improved after the revision and I recommend acceptance of the manuscript in nature communications.

Dear editors and reviewers,

Thank you very much for your positive feedback on our manuscript entitled "**(Silk Fibroin Hydrogel Adhesive Enables Sealed-tight Reconstruction of Meniscus Tears)**" (NCOMMS-23-47640B). Based on these valuable comments and suggestions, the manuscript has indeed made significant progress during the revision process. Further, we have carefully edited the language to improve its readability, making it more accessible and engaging for a wider audience.

Thank you very much for your work and help.

Sincerely,

Hongwei Ouyang, M.D., Ph.D.
Zhejiang University, Hangzhou 310058, China

Point-by-point responses to comments from the reviewers:

Reviewer #1

The authors have thoroughly and adequately answered to each question. The paper could thus be published as it is.

Response: We thank the reviewer for carefully reading the manuscript and for providing positive and very valuable feedback. We are grateful for their acknowledgment of our efforts in revising the manuscript to address all concerns.

Reviewer #2

Thanks to the authors for addressing the concerns, please carefully check out the manuscript for any grammar error.

Response: We thank the reviewer for the positive response and all valuable suggestions in previous comments to help us improve our manuscript. We have invited a native English speaker colleague to assist with the language editing of our manuscript to meet the requirements of Nature Communications.

Reviewer #3

The authors have comprehensively revised the manuscript with special focus and adding sections on the specific novelties of the current manuscript. In addition, the authors have responded fully to all my comments and other reviewers' comments and questions and added them to the manuscript.

I believe the manuscript is significantly improved after the revision and I recommend acceptance of the manuscript in nature communications.

Response: We greatly appreciate the valuable time and insightful comments from the reviewer, which have significantly enhanced the quality of our work. Furthermore, we deeply appreciate the reviewer's recommendation, and we are honored to have the opportunity to publish our work in Nature Communications.